# Structural heterogeneity of the ion and lipid channel TMEM16F

Zhongjie Ye [1,2,13], Nicola Galvanetto [3,4,13], Leonardo Puppulin[5,6], Simone Pifferi[1,7], Holger Flechsig[6], Melanie Arndt [4], Cesar Adolfo Sánchez Triviño[1], Michael Di Palma [7], Shifeng Guo[8,9], Horst Vogel [2,10], Anna Menini [1], Clemens M. Franz [6], Vincent Torre[1,11,12] ✉ & Arin Marchesi[6,7] ✉

Transmembrane protein 16 F (TMEM16F) is a $Ca^{2+}$-activated homodimer which functions as an ion channel and a phospholipid scramblase. Despite the availability of several TMEM16F cryogenic electron microscopy (cryo-EM) structures, the mechanism of activation and substrate translocation remains controversial, possibly due to restrictions in the accessible protein conformational space. In this study, we use atomic force microscopy under physiological conditions to reveal a range of structurally and mechanically diverse TMEM16F assemblies, characterized by variable inter-subunit dimerization interfaces and protomer orientations, which have escaped prior cryo-EM studies. Furthermore, we find that $Ca^{2+}$-induced activation is associated to stepwise changes in the pore region that affect the mechanical properties of transmembrane helices TM3, TM4 and TM6. Our direct observation of membrane remodelling in response to $Ca^{2+}$ binding along with additional electrophysiological analysis, relate this structural multiplicity of TMEM16F to lipid and ion permeation processes. These results thus demonstrate how conformational heterogeneity of TMEM16F directly contributes to its diverse physiological functions.

Lipid scramblases allow the passive movement of phospholipids between the two membrane leaflets, thereby reducing lipid asymmetry, altering bilayer physical properties, and orchestrating various signaling cascades[1,2]. TMEM16F belongs to the TMEM16 protein family and functions as both, a $Ca^{2+}$-activated channel as well as a $Ca^{2+}$-activated lipid scramblase[3,4]. It has been implicated in several physiological processes, including blood coagulation, bone development, membrane repair, and viral entry[5–13]. Thus, elucidating how this protein operates is of great physiological and clinical significance. Like other members of the family, TMEM16F assembles into homodimers and has

[1]International School for Advanced Studies (SISSA), 34136 Trieste, Italy. [2]Shenzhen Institute of Advanced Technology, Chinese Academy of Sciences, 518055 Shenzhen, China. [3]Department of Physics, University of Zurich, 8057 Zurich, Switzerland. [4]Department of Biochemistry, University of Zurich, 8057 Zurich, Switzerland. [5]Department of Molecular Sciences and Nanosystems, Ca' Foscari University of Venice, I-30172 Mestre, Venice, Italy. [6]WPI Nano Life Science Institute, Kanazawa University, Kakuma-machi, 920-1192 Kanazawa, Japan. [7]Department of Experimental and Clinical Medicine, Università Politecnica delle Marche, 60126 Ancona, Italy. [8]Shenzhen Key Laboratory of Smart Sensing and Intelligent Systems, Shenzhen Institute of Advanced Technology, Chinese Academy of Sciences, Shenzhen 518055, China. [9]Guangdong Provincial Key Lab of Robotics and Intelligent System, Shenzhen Institute of Advanced Technology, Chinese Academy of Sciences, Shenzhen 518055, China. [10]Institut des Sciences et Ingénierie Chimiques (ISIC), Ecole Polytechnique Fédérale de Lausanne (EPFL), Lausanne, Switzerland. [11]Institute of Materials (ION-CNR), Area Science Park, Basovizza, 34149 Trieste, Italy. [12]BIoValley Investments System and Solutions (BISS), 34148 Trieste, Italy. [13]These authors contributed equally: Zhongjie Ye, Nicola Galvanetto. ✉e-mail: torre@sissa.it; a.marchesi@staff.univpm.it

a double-barreled architecture (Supplementary Fig. S1a)[14–16]. Each monomer has 10 transmembrane helices (referred as TM1-TM10) and cytoplasmatic N- and C-termini connected to TM1 and TM10, respectively (Fig. 1a)[14,16]. An independent permeation pathway within each protomer is formed by TM3-TM7 and is known as the subunit cavity (Fig. 1a, b and Supplementary Fig. S1b). In the TMEM16 family, opening and closing of the cavity are controlled by two primary $Ca^{2+}$-binding sites located between TM6-TM8 and, in some mammal family members like mouse TMEM16F, by an additional regulatory $Ca^{2+}$-binding site within TM2 and TM10 (Fig. 1b and Supplementary Fig. S1a)[14,17,18]. As observed in several other ion channels, the structure of the subunit cavity is reminiscent of an hourglass[19].

Current activation models originated from fungal scramblase propose that upon calcium binding separation of TM4 and TM6 helices exposes a membrane-spanning hydrophilic furrow to the hydrophobic stratum of the bilayer, destabilizing the membrane and lowering the energy barrier for lipid transverse translocation (Fig. 1b)[20–23]. Lipid transport across bilayer would proceed according to the canonical in-groove "credit card" model, whereby polar lipid head groups inhabit the membrane-facing hydrophilic furrow while their hydrophobic tails remain anchored in the bilayer core[3,20,24,25]. Intermediate states with an intact TM4/TM6 interface and a dilated protein enclosed conduit would be, akin to TMEM16A/B active conformations, scrambling-incompetent but permissive to ionic permeation[20,22,26–28]. This model, which has sometimes been referred to as alternating pore/cavity mechanism[3,14,15,25], implies the existence of discrete conformations in thermodynamic equilibrium promoting either ion conduction or scrambling (Fig. 1b). Structural investigations based on single particle imaging obtained with cryo-EM microscopy have provided several TMEM16F molecular structures both in the absence or presence of $Ca^{2+}$ (refs. 14,16,29). Nonetheless, in most cases, the lipidic and ionic permeation pathways appear to have been trapped in an inactive or closed conformation[3,15] wherein the subunit cavity only shows $Ca^{2+}$-induced bending of the intracellular portion of TM6 (Supplementary Fig. S1b). Only very recently, major rearrangements of helices TM3 and TM4 ensuing partial opening of the subunit cavity were evidenced in mutants exhibiting an activating phenotype, suggesting a possible structural bias toward inactive conformations in previous cryo-EM studies[30].

Thereby, it remains unresolved, and a matter of debate whether the lack of structural evidence in support of the alternating pore/cavity model stems from intrinsic limitations of cryo-EM technology or is attributable to alternative ion and lipid permeation mechanisms, which have also been proposed[16,29–33]. Indeed, one has to consider that in cryo-EM studies particle selection and intensive 2/3D classification procedures discard the majority of single particle images[34], suggesting a bias towards a predominant and thermodynamically more favorable molecular structure[35]. Moreover, to which extent the properties of lipid nanodiscs, the membrane mimetic system of choice in structural studies, are similar to native extended membrane bilayers remains debated[36]. For instance, it has been shown that the lipid bilayer within nanodiscs is strongly anisotropic and inhomogeneous[37] and that lipid confinement strongly affects membrane elastic properties[38] whereby the latter regulates the structure–function relationships of several membrane proteins, including TMEM16s[39,40]. Accordingly, the conformational energy landscape of embedded proteins might well be affected in ways that are difficult to predict when confined within nanodiscs, further complicating the functional annotation of solved structures. Thus, it is not surprising that the $Ca^{2+}$-bound structures of TMEM16A and TMEM16F were captured in non-conductive or intermediate states despite the presence of ligands and substrates in high concentrations[14,16,28,41].

To tackle these issues and gain deeper insights into the conformational trajectory and functioning of TMEM16F, in this work, we investigate TMEM16F mechanical, structural, and dynamic properties

by employing two powerful single-molecule methods based upon atomic force microscopy (AFM) technology, namely single-molecule force spectroscopy (SMFS) and high-speed AFM (HS-AFM) imaging[42]. Both techniques have proven successful in previous studies, including our own, in characterizing conformational changes of membrane proteins under physiologically relevant buffer conditions, temperature, and membrane composition[43–51]. HS-AFM and SMFS provide complementary and synergistic information: SMFS measures the mechanical properties of intra- and intermolecular interactions that stabilize membrane proteins and allocate them to structural regions[52,53]; HS-AFM allows real-time observation of single molecule surface structure and dynamics with high lateral (~1 nm), vertical (~0.1 nm), and temporal (~100 ms) resolution[54].

Here we show that TMEM16F structure explores a range of distinct conformations involving drastic changes in the dimerization interface and interaction with the surrounding bilayer, which have escaped prior cryo-EM studies. Furthermore, our SMFS experiments indicate that $Ca^{2+}$ binding and TMEM16F activation are associated with rearrangements of the TM3, TM4, and TM6 helices, reminiscent of those previously described for fungal and mouse TMEM16F scramblase. Finally, the analysis of several electrophysiological quantities provides a straightforward connection between the observed structural multiplicity and the variability in functional properties. Thus, our results are indicative of a shallow energy landscape, leading to a hitherto overlooked conformational heterogeneity possibly facilitating ion and lipid movement across cell membrane. We propose that this structural multiplicity establishes the physical foundation for a tight spatial and temporal control of TMEM16F's diverse physiological functions.

## Results
### Mechanical unfolding of native and recombinant TMEM16F
To investigate the mechanical properties of TMEM16F and its interaction with the surrounding lipids, we performed SMFS experiments on isolated cell membrane fragments. We previously developed an unroofing pipeline to collect plasma membranes, exposing the intracellular leaflet devoid of cytosolic-soluble components to the AFM probe[55,56]. This method proved to be highly efficient and reproducible in harvesting membranes from neuroblastoma NG108-15 cells compared to other commonly used cellular strains. Furthermore, we show that TMEM16F (WT-16F) is widely expressed on the membrane of these hybrid cells (Supplementary Figs. S2a, S3, and S4) where its unfolding spectra can be reliably identified[56]. Consequently, NG108-15 cells provide a convenient platform for characterizing TMEM16F in its native environment, and therefore this pair was chosen for further SMFS analysis. In a typical experiment, the apical membranes exposing the cytosolic face to the AFM stylus were first isolated by mechanical means (see Methods, and Supplementary Fig. S2b, c)[55,56]. Next, single-layered membrane patches were identified by AFM imaging (Fig. 1c and Supplementary Fig. S2d, e) and studied by SMFS. Unspecific physisorption of the membrane proteins to the AFM stylus was attained by gently pushing the AFM tip onto the sample (contact force 1 nN) (Fig. 1d, left panel). After a contact time of 0.6 s, the probe was retracted while simultaneously measuring the force necessary to unfold/stretch the protein and the distance traveled by the tip (Fig. 1d, mid and right panels). This procedure was repeated ~2 million-times and the unfolding spectra collected - also referred to as force-distance (F-D) curves - were screened and clustered by an unsupervised clustering procedure which we have previously developed and validated[56,57] (see "Methods" section). To further aid identification of TMEM16F F-D curves, we also overexpressed the fusion construct His$_6$-N2B-TMEM16F-GFP (N-N2B-16F), which incorporates a hexahistidine tag followed by N2B segment (a 204 a.a. long fingerprint commonly used in SMFS experiments[47]) at the N-terminal, and the green fluorescent protein (GFP) at the C-terminal end (Fig. 1e, left panel). This construct is easily recognized from the collected F-D traces because of

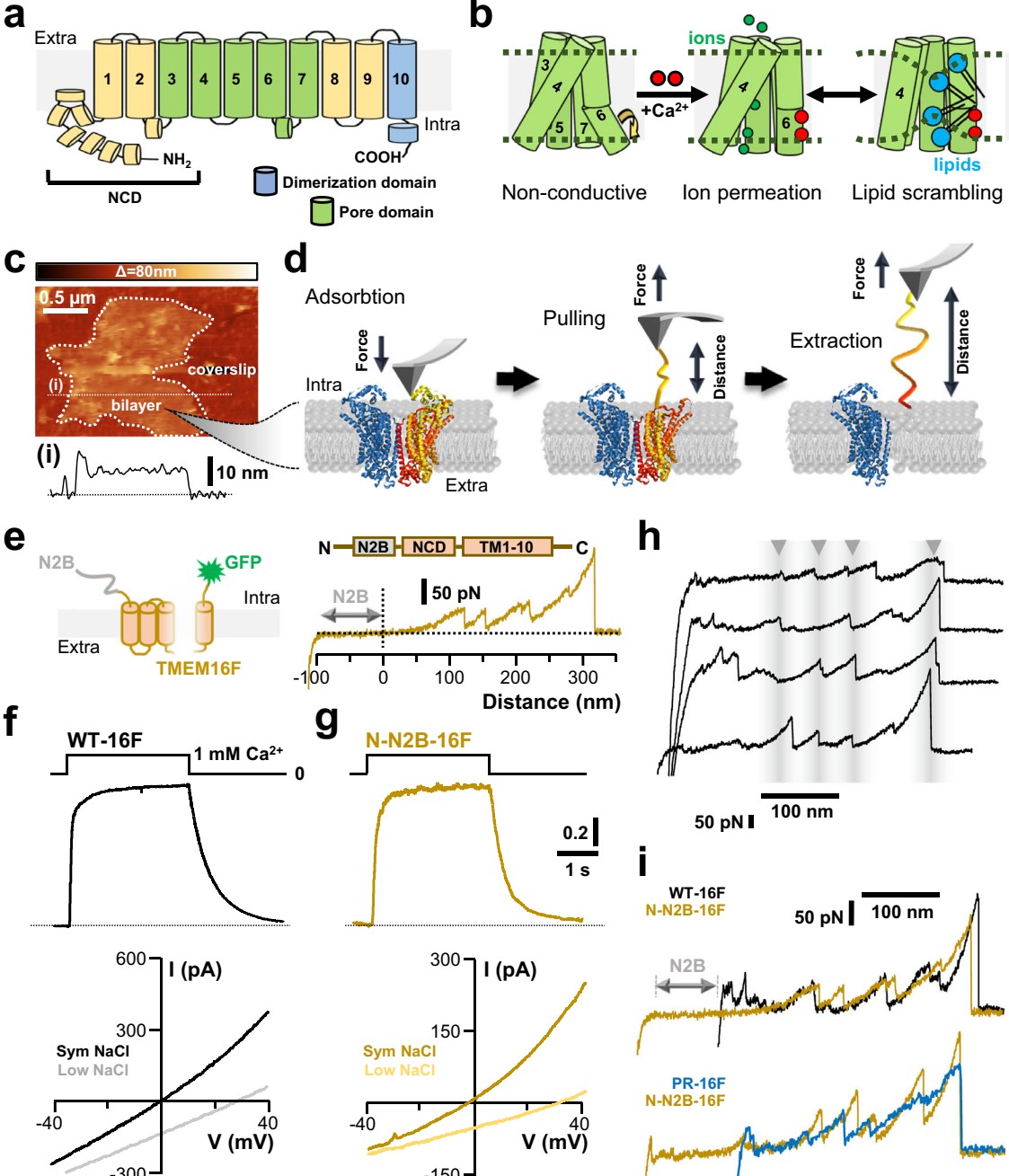

**Fig. 1 | Native and recombinant TMEM16F have similar unfolding and electrical properties. a** Membrane topology of a TMEM16F subunit. The transmembrane helices (numbered from 1 to 10) important for ion/lipid translocation and subunits dimerization are highlighted in green (pore domain) and sky-blue (dimerization domain), respectively. NCD, N-terminal cytosolic domain. **b** Cartoon showing the pore domain architecture of TMEM16F. The permeation pathway is gated by $Ca^{2+}$ ions (red-filled circles) and can function as an ion channel (middle panel), and/or lipid scramblase (right panel). **c** Membrane patches isolated from NG108-15 cells (dotted outline) were identified by AFM imaging ($n = 47$ independent topographic images) before unfolding experiments commenced. Cross-sectional analysis along the white dotted line is shown in **i**. **d** Schematics illustrating the unfolding process of a TMEM16F protomer. The AFM probe is first brought into contact with a TMEM16F to aid physisorption of the NCD to the AFM tip. Afterwards, the probe is retracted, thereby applying a mechanical force unfolding the protein, and an F-D curve is recorded. **e** Recombinant TMEM16F was engineered to bear the N2B

fingerprint and the GFP polypeptide at the N- and C-terminus, respectively (left panel). A representative unfolding of the N-N2B-16F construct from the N-terminus is shown at the right. The double-headed arrow designates the unfolding of the N2B segment. The C-terminal GFP allows individuation of successfully transfected cells and location of TMEM16F on the cell membrane. **f** Representative normalized currents recorded from inside-out membrane patches from HEK293 cells expressing WT-16F. The upper panel shows currents recorded at the holding potential of +60 mV upon a $Ca^{2+}$ concentration jump from nominally 0 to 1 mM. The lower panel shows currents activated by voltage ramps from −40 mV to +40 mV in the presence of 1 mM $CaCl_2$ in symmetrical 140 mM NaCl (Sym NaCl) or 14 mM intracellular NaCl solutions (Low NaCl). **g** As in **f** but for the N-N2B-16F fusion construct. **h** Examples of F-D spectra corresponding to the mechanical unfolding of WT-16F from the N-terminus in the absence of $Ca^{2+}$. Arrowheads designate commonly observed unfolding intermediates. **i** Superposition of representative WT-16F (black), N-N2B-16F (ochre), and purified TMEM16F (blue lines) reconstituted in proteoliposomes.

the N2B signature, which is characterized as a non-unfolding event[58]: the N2B segment is stretched with a force lower than 20 pN and a contour length (Lc) of about 85 nm[47]. For this reason, F-D traces which at the beginning have a flat ~85 nm long segment followed by an unfolding pattern extending over a distance approximating the fully stretched TMEM16F sequence of 870-911 a.a., - i.e. 340–365 nm, assuming the length of a single a.a. is 0.4 nm - are recognized as the N-N2B-16F construct unfolding from the N-terminal end (Fig. 1e, right panel). Thus, this fusion protein provides a reliable unfolding template for TMEM16F, unambiguously identifying TMEM16F unfolding polarity and distinguishing it from TMEM16B channels, which are also expressed on NG108-15 membranes (Supplementary Fig. S2a and S3) and have a similar amino acid sequence (Supplementary Fig. S12) and gross 3-D architecture[15,25]. Furthermore, the functionality of the N-N2B-16F construct was tested in HEK293 cell lines by patch-clamp recordings in excised inside-out configuration (Fig. 1f). Electrical measurements showed that WT-16F and N-N2B-16F have very similar $Ca^{2+}$-dependent currents and selectivity toward cations (Fig. 1f, g), indicating that $His_6$-N2B and the GFP polypeptides did not affect TMEM16F function.

The unfolding of membrane proteins follows some principles which guide the interpretation of F-D curves obtained by SMFS. If the membrane protein has n transmembrane segments $TM_i$ i = 1……n, their unfolding occurs sequentially: when the cantilever tip is attached to the N-terminus, the first transmembrane segment to be unfolded is TM1 followed by TM2 and so on[46–48,50,53]. If the N-terminal is connected to TM1 by a large structured cytoplasmic domain (as in the case of the TMEM16F, see Supplementary Fig. S1, S12, and Supplementary Table 1), the unfolding of its elements will not be sequential but instead display mechanical hierarchy: the domains requiring a lower unfolding force will be the first to be unfolded[58,59]. The F-D traces note the steps of this unfolding process in a sequence of force peaks. Representative unfolding spectra from endogenous (Fig. 1h) and recombinant TMEM16F bearing the N2B fingerprint (Fig. 1i) display the characteristic saw-tooth pattern, which has been previously reported for many membrane and soluble proteins[47,48,52,53,56,58]. Despite some variability, similarities in the pattern of force peaks across different unfolding events were observed for both, WT-16F and N-N2B-16F, with peaks varying much in amplitude but less in distance, indicating the presence of obligatory unfolding intermediates (Fig. 1h, i).

## TMEM16F mechanical and structural properties are heterogeneous

Because the unfolding process and the occurrence of force peaks are stochastic in nature, 78 of F-D spectra from N2B-tagged TMEM16F obtained in identical conditions were aligned, superimposed, and displayed as density plots (Fig. 2a). Statistical analysis of these curves indicates that the obtained F-D traces from $Ca^{2+}$-free solution exhibit characteristic force peaks with a similar Lc value (see further on), but with rupture forces varying from 20 up to 250 pN. The magnitude of each force peak is a proxy of the interactions stabilizing the different unfolded structural units[52,53]. Thus, these differences point to variable intra-molecular interactions within each unfolded segment and/or the surrounding membrane environment. To better characterize this heterogeneity, we computed the unfolding work (W) necessary to pull these proteins[60]. The corresponding violin plot has an average unfolding W of $16.4 \times 10^{-18}$ J (aJ) and a bimodal distribution with well-resolved peaks around 11 and 22 aJ (Fig. 2b, left panel). Similar unfolding patterns of TMEM16F displaying characteristic force peaks with variable rupture forces were also observed in non-transfected cells (WT-16F), and artificial bilayer membranes (PR-16F) where purified and

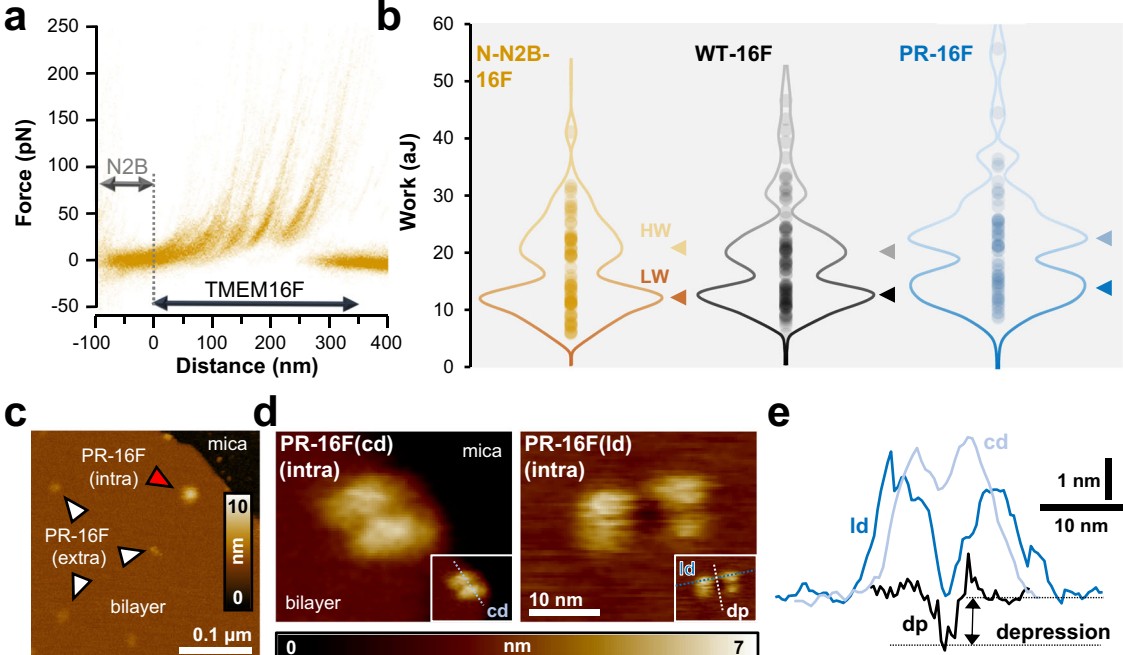

**Fig. 2 | Mechanical and structural features of TMEM16F. a** Superposition of 78 F-D curves from recombinant N-N2B-16F pulled in the absence of $Ca^{2+}$. Only the unfolding of TMEM16F monomers is shown. **b** Violin plots of the calculated unfolding work for the N-N2B-16F (n = 78; ochre hue), WT-16F (n = 101; gray hue) and purified murine TMEM16F reconstituted into liposomes (PR-16F, n = 68; blue hue). Average work and standard deviations correspond to $16.4 \pm 7.5$, $18.4 \pm 8.4$, and $20.4 \pm 8.7$ aJ for N-N2B-16F, WT-16F, and PR-16F constructs, respectively. A multimodal distribution with two prominent populations referred to as low (LW) and high unfolding work (HW) was observed in all constructs (arrowheads). **c** Medium-resolution AFM image of reconstituted TMEM16F protein (n = 4 independent AFM experiments). Individual TMEM16F exposing either extracellular or intracellular side are clearly discernible (white and red arrowheads, respectively). **d** High-resolution images of two representative TMEM16F dimers exposing the intracellular face. Compact (PR-16F(cd), left; n = 5 independent molecules/experiments) and loose (PR16F(ld), right; n = 5 independent molecules/experiments) configurations with the two protomers further apart were observed. Both molecules were imaged within 24 h with the same HS-AFM scanner, and image area dimensions. **e** Height profile analysis along the white (dp) and blue (cd and ld) dashed lines (see insets). In the loose configuration, a ~1 nm membrane depression is often observed in-between the TMEM16F protomers.

reconstituted TMEM16F was the sole protein present (Fig. 1i and Supplementary Fig S5). In both cases, inspection of W distributions reveals the presence of two dominant populations (Fig. 2b, arrowheads) indicating that the observed heterogeneity does not arise from spurious unfolding events, such as heterodimers between endogenous TMEM16F/B proteins or the overexpressed N-N2B-16F construct. The F-D curves were sorted accordingly in low and high unfolding W (Supplementary Fig. S6a, dark and pale ochre hue, respectively), both of which had similar force peak locations (Supplementary Fig. S6b). Furthermore, some F−D curves were longer and exhibited an additional force peak with a maximal contour length (Lcmax) of ~370–390 nm (Supplementary Fig. S6), roughly corresponding to the unfolding of the whole TMEM16F (i.e. 911 a.a.). This behavior is reminiscent of the CNGA1 channel, which also has an additional peak when the full polypeptide is pulled out of the membrane[47]. Hence, unfolding of TMEM16F in the absence of $Ca^{2+}$ indicates the existence of two major sources of structural heterogeneity: first, monomers which can be unfolded with a low and high W (Fig. 2b) and second, monomers which had variable anchoring values of Lcmax indicating different detachments at the C-terminal end (Supplementary Fig. S6).

To gain further insight into TMEM16F assembly and interaction with the phospholipid membrane, we performed HS-AFM imaging experiments. Because AFM operated in imaging mode has a low discriminatory power when applied to native, protein-crowded membrane fragments, reconstituted TMEM16F was studied. This preparation has previously been demonstrated to retain $Ca^{2+}$-dependent lipid transport activity[14]. Proteoliposomes were adsorbed onto freshly cleaved mica and subjected to HS-AFM imaging in a physiological buffer solution (see Methods). Large membrane areas of ~5 nm thickness embedding TMEM16F proteins were readily identified (Fig. 2c). At medium resolution, TMEM16F appear as unstructured protrusions bulging from the planar bilayer. Based on the height analysis, TMEM16F displayed two distinct populations providing height differences of ~1.6 nm and ~3.8 nm from the bare bilayer (Fig. 2c and Supplementary Fig. 1c). A close inspection of TMEM16 cryo-EM structures suggested similarly large differences in height for the extracellular and intracellular faces (Supplementary Fig. S1a), thus guiding sidedness assignment: molecules protruding from the membrane by about 1.5 and 3.5 nm were deemed to represent TMEM16F from the extracellular and intracellular side, respectively. Imaging individual TMEM16F from the cytosolic face at high resolution revealed the expected dimeric assembly of the protein, where individual subunits could be contoured at high definition (Fig. 2d). Interestingly, these high-resolution topographies suggested the existence of two major classes of TMEM16F dimers: compact (Fig. 2d, left panel) and loose dimers featuring membrane distortion (Fig. 2d, right panel). While compact dimers have a full width at half maximum (FWHM) of around ~11 nm, and their morphology appears overall in agreement with the expected shape reported in cryo-EM studies[14,16] (Fig. 2d, e and Supplementary Fig. S1a), loose dimers are larger, with a FWHM of around ~16 nm and often feature a characteristic membrane depression of ~1 nm around the dimer interface (Fig. 2d, e). Noteworthily, membrane distortion and thinning were evidenced in several TMEM16 scramblase structures, including but not limited to TMEM16F[16,20,22]. Therefore, we suggest that these dimers with different geometrical properties correspond to the two classes of F-D traces denoting unfolding with a low and high W. We furthermore speculate that a high W was associated with the unfolding of compact dimers owing to a stronger inter-subunit coupling and interaction forces with the bilayer, although correlative force spectroscopy and imaging experiments from the same specimens will be needed to confirm this hypothesis.

## TMEM16F dimer interface is dynamic

Before the first cryo-EM structures were released, it has been proposed that dimerization of TMEM16A and TMEM16F polypeptides occurs through cytosolic N-terminal interactions[61]. However, this observation is not supported by existing cryo-EM structures of TMEM16A and TMEM16F, where a back-to-back arrangement of the protomers favors oligomerization via their C-terminal TM10 (Supplementary Fig. S1a)[14,16,28,41]. These contrasting reports and the observation of structurally and mechanically diverse TMEM16F assemblies (Fig. 2) guided us to further investigate how monomers interact.

Inspection of our SMFS dataset obtained from NG108-15 cell lines overexpressing the N-N2B-16F construct identified F-D traces with a Lcmax of about 500-700 nm, i.e. roughly corresponding to twice the Lcmax value of a TMEM16F monomer. These longer traces not only harbor the N2B fingerprint either at the beginning or halfway into the spectra (Supplementary Fig. S7a−c), but also bear many force peaks consistent with those observed in TMEM16F monomers. Akin traces of about 600 nm were also collected from neuroblastoma cell lines expressing endogenous WT-16F (Fig. 3a) as well as proteoliposomes (Supplementary Fig S7a), ruling out that the introduced GFP and N2B tags might have caused alien interactions. We therefore reasoned that due to the strong mechanical coupling between the monomers, TMEM16F might occasionally be unfolded in tandem, as the concatenation of two monomers.

To prove the tandem arrangement and better dissect the TMEM16F dimer inter-subunit interactions, we constructed a scramblase bearing the N2B fingerprint attached to the C-terminal end (GFP-TMEM16F-N2B-His₆). Whilst the N-N2B-16F construct provides a template for the N-terminal unfolding, the C-terminal N2B-tagged TMEM16F (C-N2B-16F) establishes a reference with opposite polarity, where TMEM16F unfolds from TM10 on towards TM1. Indeed, if the longer identified spectra correspond to the extraction of TMEM16F dimers, the concatenation of two TMEM16F monomers unfolding from either N- or C-terminus will overlay well and provide a gross identification of the domains important for dimerization. Based on these two reference constructs, we identified three types of basic combination or dimer models, dubbed C-C, C-N, and N-N (Fig. 3a−d and Fig. S7a−c). In the C-C model (Fig. 3b, d), the N-N2B-16F and the C-N2B-16F templates (red and blue trace, respectively) could be aligned in sequence - one after the other - to the unfolded dimers of native WT-16F. Thus, we interpreted the unfolded pattern as two adjoined TMEM16F polypeptides pulled from the N- and the C-terminal end, which were linked together via C-terminal interactions (Fig. 3d). This unfolding configuration was also found in recombinant TMEM16F dimers bearing the N2B fingerprint (Supplementary Fig. S7a) and is reminiscent of the cryo-EM structures, where protomers interaction is mediated by TM10 helices. Besides the C-C assembly, we also observed dimers which could be superimposed to either two N-N2B-16F templates (Supplementary Fig. S7b) or to the C-N2B-16F and N-N2B-16F constructs in sequence (Fig. 3c and Supplementary Fig. S7c). We therefore inferred that TMEM16F subunits additionally associate through heterotypic interactions involving the C- and N-terminal domains (Fig. 3d, C-N model) or homotypic interactions mediated by the NCDs (Fig. 3d, N-N model).

It is noteworthy that occasionally we unfolded putative TMEM16F trimers (Supplementary Fig. S7d), which were also visualized by high-resolution AFM imaging (Supplementary Fig. S7e). Although these observations were rare and their biological significance unclear, they constitute additional evidence of variable oligomerization interfaces. Interestingly, non-canonical assemblies were recently reported for TRPV3, a non-related channel belonging to the voltage-gated ion channel superfamily, which was observed in an unusual pentameric state[62].

Prompted by this SMFS finding, we sought to further capture and demonstrate these alternative quaternary arrangements by HS-AFM imaging. To this end, we imaged single TMEM16F exposing the cytoplasmic side to the AFM probe in similar buffer conditions to SMFS experiments (see Methods). Protomers rotational and translational

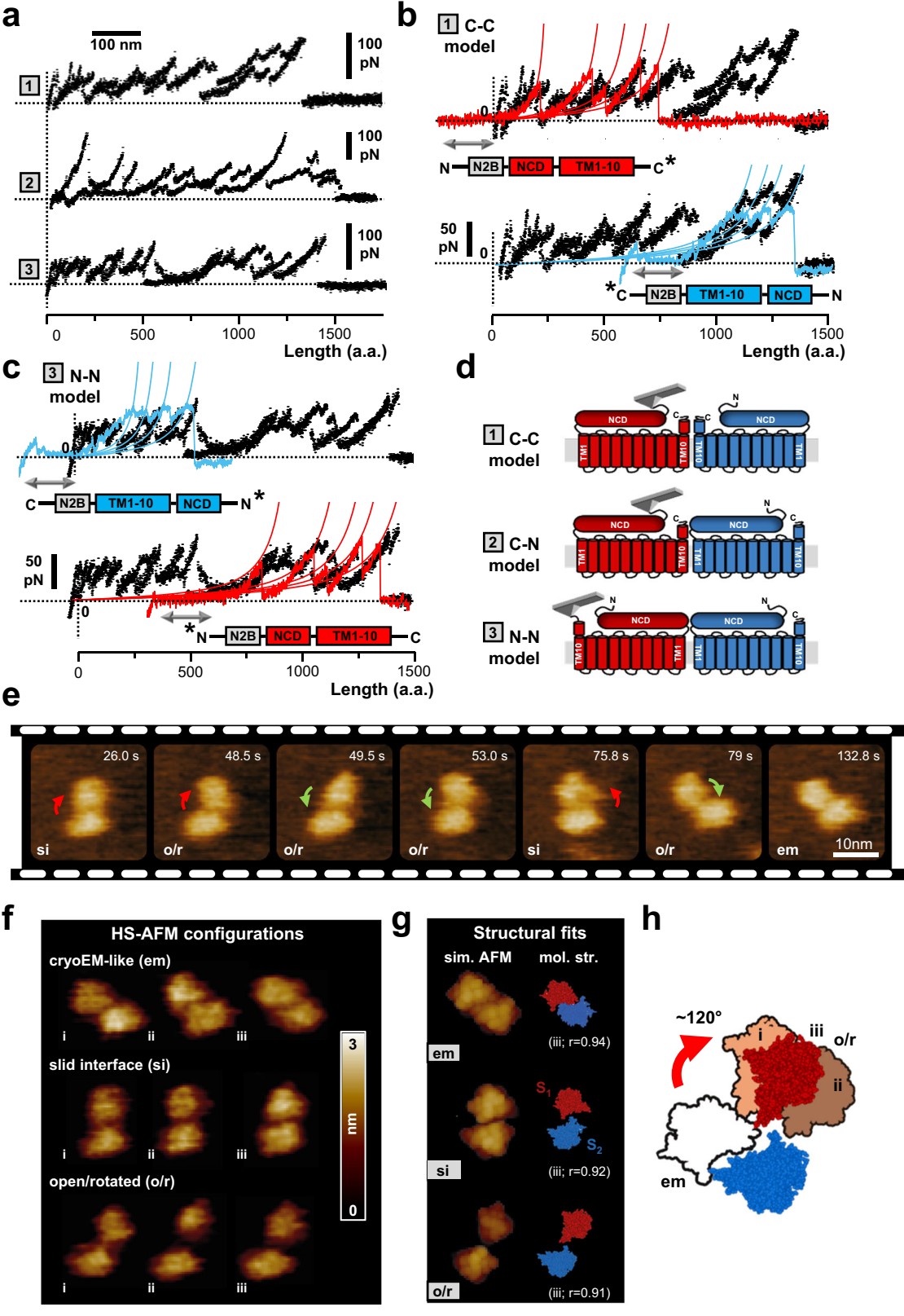

movements were observed, demonstrating that the protein quaternary structure is indeed dynamic (Fig. 3e and Supplementary Video 1). Real-time imaging shows that the two subunits swing away and back amid compact-symmetric and loose-asymmetric arrangements, similar to those described in Fig. 2d. At the beginning of the video, the dimer adopts a compact configuration, where the two subunits appear symmetrically coordinated (Fig. 3e, t = 26.0 s).

However, after a few seconds, the upper subunit disengages and gradually rotates away from the symmetry axis (Fig. 3e, t = 48.5 s, 49.5 s). The dimer adopted a loose and asymmetric arrangement for about 40% of the recording, and toward the end of the sequence (Fig. 3e, t = 132.8 s), TMEM16F reverts back to a more compact and symmetric configuration. Fitting of the cryo-EM TMEM16F dimer into HS-AFM images[63,64] indicates a remarkably high agreement between the 3D

**Fig. 3 | TMEM16F has a dynamic inter-subunit interface. a** Endogenous TMEM16F channel occasionally unfolds in tandem. Three main unfolding patterns (labeled 1–3) were observed, revealing differences in inter-subunits mechanical interactions. **b, c** The hypothesized dimerization models (C-C, C-N, and N-N) were identified by concatenating to WT-16F unfolded dimers (black traces) representative spectra from TMEM16F monomers wherein the N2B tag was conjugated either to the N- (N-N2B-16F in red colors) or C-terminal end (C-N2B-16F in cyan colors). Sketches of the engineered constructs and their putative unfolding polarity are shown at the bottom of the unfolded spectra. (*) denotes the inferred dimerization interface. Gray double-headed arrow designates the unfolding of the N2B segment. **d** Schematics depicting the three proposed dimerization models. In C-C model protomers interaction is mediated by C-terminal domains (1); in C-N model by C- and N-terminal domains (2); and in N-N model by N-terminal domains (3). Red and blue colors denote the two TMEM16F subunits. **e** HS-AFM images (from Supplementary Movie 1) of a TMEM16F dimer from the intracellular side. Monomers display significant relative motion, swinging away (red arrows) and back (green arrows) amid compact-symmetric dimers and loose-asymmetric arrangements. Similar subunit motions (sliding and/or rotation, see main text) were observed in three independent experiments. **f** The observed dimer configurations can be qualitatively sorted into three structural classes referred to as cryo-EM like (em), slid (si), and open/rotated (o/r). Three examples for each class (i–iii) are reported. **g** Simulated AFM and automatized fitting procedures were used to reconstruct the TMEM16F quaternary structure from the deposited TMEM16F cryo-EM file (pdb 6P46, see Methods). Simulated AFM images (left) and corresponding molecular structures (right) obtained after fitting are shown for the three main structural classes (em, si, o/r). Similarity scores (r) are reported. $S_1$ (red) and $S_2$ (blue) designate the two protomers. **h** Cryo-EM-like (em) and open/rotated (o/r i–iii) configurations were superimposed to highlight rotational motions of subunits. Subunit $S_2$ (in blue) was used as a reference for structures registration. A clam-shell mechanism describes sufficiently well the observed conformational dynamics. Em and o/r structural classes are filled in white and different red shades colors, respectively. The o/r class shows some structural fluctuations (compare i–iii).

high-resolution structure and the surface topography observed at $t = 132.8$ s, with a correlation score of 0.94 (Fig. 3f, g). However, we could not reproduce sufficiently well the other TMEM16F morphologies shown in Fig. 3e from any of the reported cryo-EM structures deposited thus far. Visual inspection of the HS-AFM movie sequence suggests that the rather heterogenous structural ensemble observed could be grouped into three distinct structural subsets or classes, which we refer to as cryo-EM-like (Fig. 3e–g; em), slid (Fig. 3e–g; si), and open/rotated (Fig. 3e–g; o/r). To better understand the assembly changes, instead of docking the whole dimeric TMEM16F structure, we separately fitted each TMEM16F protomer into the AFM contoured images. A combined structural model of the dimeric assembly with altered domain arrangements was thus obtained (Fig. 3g, see Methods). This analysis gave correlation scores >0.9 for all the observed conformations and are suggestive of two types of motion between the protomers: (1) a ~ 5 nm outward glide along the symmetry axis resulting in the slid configuration (compare em and si subsets in Fig. 3f, g); (2) a clam-shell rotation pivoting around the membrane-facing surface of the two subunits, leading to the open/rotated states (Fig. 3f, g, o/r subset). Structural comparison of the hybrid cryo-EM/AFM structural models locates the dimerization interfaces toward the intracellular end of TM10 and C-terminal cytoplasmic helices in the slid configuration, and TM3 of one and TM10 of the other subunit in the open/rotated states. Some structural flexibility was also observed in the open/rotated cluster (compare Fig. 3e at $t = 48.5$ and 49.5 s), suggesting that the dimerization interface might further extend to encompass the bulky NCD (Fig. 3h, i and ii).

Taken together, SMFS and HS-AFM results point toward conformational heterogeneity in TMEM16F, where large changes in the dimerization interface led to a diverse and dynamic structural ensemble. These findings corroborate previous cryo-EM[14] and co-immunoprecipitation[61] data, suggesting that TMEM16F might dimerize through either C- or N-terminal regions, and hence adopt alternative quaternary structures.

### Conformational changes in TMEM16F upon Ca²⁺ binding

To identify conformational changes induced by $Ca^{2+}$ binding, we collected a total of 336 F-D traces from WT-16F and N-N2B-16F constructs and compared their unfolding in the absence ($n = 179$) and presence ($n = 157$) of saturating $Ca^{2+}$ (Fig. 4a–d and Supplementary Fig. S8). N-N2B-16F and WT-16F curves were found to superimpose well, displaying a similar unfolding pattern and force peak positions (Supplementary Fig. S8), which indicates that the N2B and GFP polypeptides did not affect the mechanical properties of TMEM16F. We therefore focused our analysis on the N-N2B-16F construct, whose dataset is less likely to be affected by false positive events (e.g. TMEM16B unfolding).

Density plots in the presence and absence of $Ca^{2+}$ reveal six well-correlated force peaks occasionally preceded by a more variable rupture force, the position of which fluctuates stochastically (Fig. 4a, c; arrowheads). Every single force peak for each individual F-D curve was fitted using the worm-like chain (WLC) model to uncover the polypeptide lengths of the stretched structural segments[46,47,50]. Representation of the obtained Lc into a histogram allowed the six most common lengths to be identified and fitted with a multiple Gaussian model, yielding mean contour length values. Whilst the first and the fifth Lc sizes in the presence and absence of ligand were found to be indistinguishable within the experimental accuracy, substantial differences were detected while stretching the intervening polypeptide segments (compare right panels in Fig. 4a, c). The second, third, and fourth unfolding peaks were found to have Lc lengths of $212 \pm 9$, $259 \pm 11$, and $288 \pm 13$ nm in the $Ca^{2+}$-free condition and $189 \pm 6$, $227 \pm 12$, and $276 \pm 10$ nm in the $Ca^{2+}$-bound condition, respectively (see also Supplementary Table 1). Albeit peak at 189 nm in the $Ca^{2+}$-bound state was observed at somewhat lower occurrences (33%, Supplementary Table 1), it was present in most spectra collected from the constitutively scrambling Y563K mutant[65] in the absence of $Ca^{2+}$ (75%; Supplementary Fig. S9), further stressing its importance for efficient lipid transport. The variations detected in the last unfolding peak (denoted by *dtc.* in Fig. 4a–d) were ascribed to the variability of the final detachment event[47,66], rather than to intrinsic physical properties of TMEM16F, and were not further considered. Thus, the different unfolding pathways between the *apo* and $Ca^{2+}$-bound states are genuine and indicative of rather drastic ligand-induced structural transitions.

To better interpret the observed mechanical changes in the context of TMEM16F 3D structure, we mapped the Lc values of the unfolded segments onto the TMEM16F primary sequence and membrane topology (Fig. 4b, d). This analysis identifies the structural units unfolded during each unfolding step and locates the $Ca^{2+}$-induced conformational changes between TM3-TM9 (marked by dashed lines in Fig. 4b, d). Specifically, two major structural transitions involving TM3 and TM6 were found. First, in the absence of $Ca^{2+}$, TM3 and TM4 unfold in pair (Lc= 212 nm) whilst upon activation, TM3 breaks-up into two segments (Lc=189 nm), with its extracellular portion remaining coupled to TM4 (Fig. 4d). Notably, the peak with Lc ~189 nm observed in the presence of $Ca^{2+}$ approximately corresponds to residue G473 (located at around 3/4 of TM3), which is a known helix breaker. This result conforms to a recent structural study that revealed key rearrangements of helices TM3/TM4 to open up the ion and lipid translocation pathway in response to $Ca^{2+}$ binding[30]. Second, in the closed state, the TM6 is mechanically coupled to TM5 (Fig. 4b, Lc = 259 nm) but upon $Ca^{2+}$ binding it unfolds in twosome with TM7 (Fig. 4d, Lc = 276 nm). These $Ca^{2+}$-induced differences are also well in agreement with cryo-EM structures which indicate a movement of TM6 toward TM7 and TM8 upon $Ca^{2+}$ binding[14,16,22].

To further define the conformational transitions, we complemented SMFS data with real-time HS-AFM imaging of single

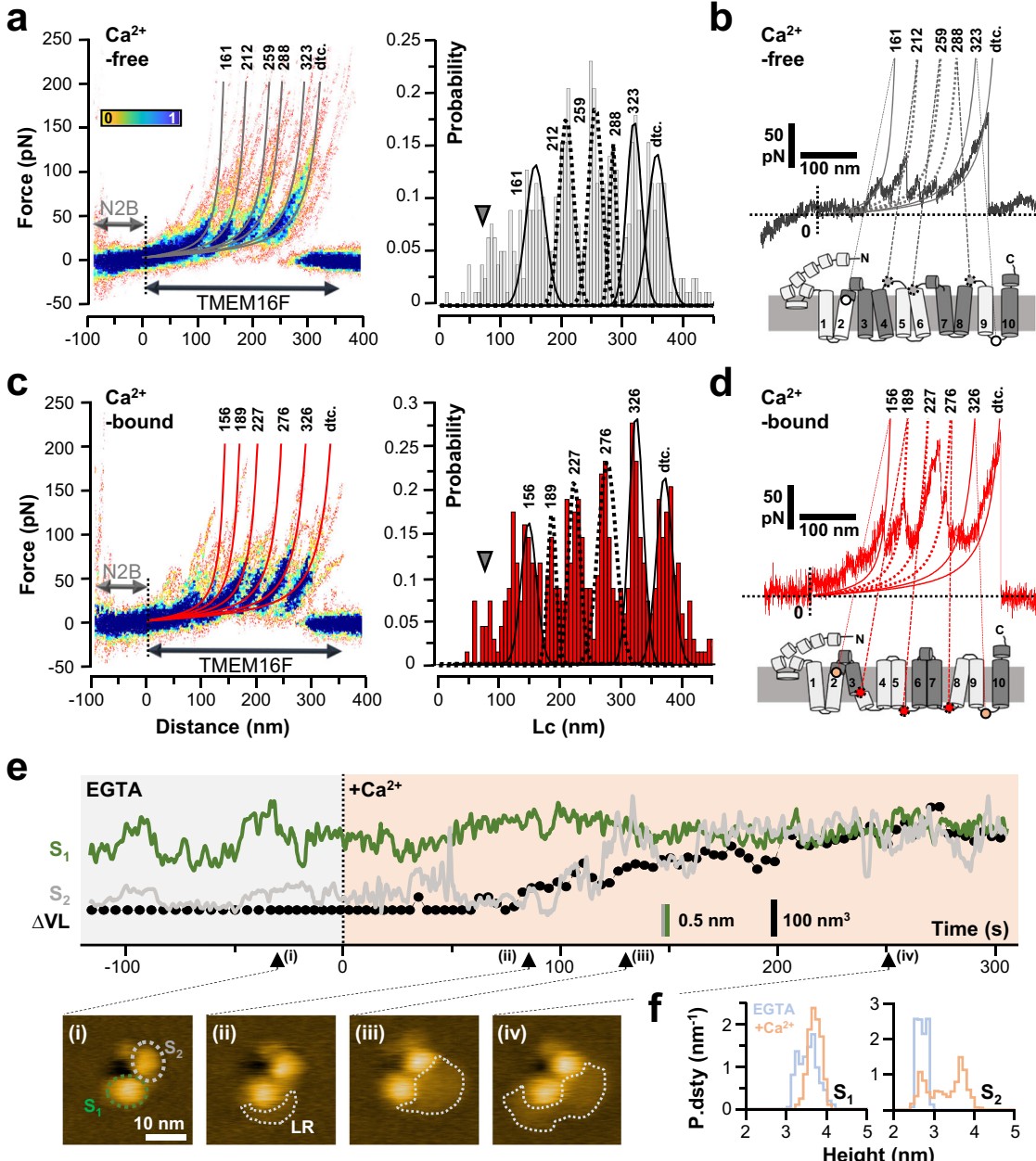

**Fig. 4 | Ca²⁺- induced conformational changes in TMEM16F. a** TMEM16F unfolding in the absence of Ca²⁺. Left panel: Superposition of 78 F-D curves from unfolding recombinant N-N2B-16F obtained in the presence of 1 mM EGTA. Worm-like chain (WLC) curves corresponding to average contour lengths (Lcs) of each force peak are overlayed (gray curves). Right panel: Contour length (Lc) histogram of all force peaks detected in the F-D curves shown in the density plot at the left. Histogram was fitted with multiple Gaussian providing mean Lcs (indicated at the top of each Gaussian distribution and WLC curve) and force peak probabilities (see text and Supplementary Table 1). The six major peak classes were occasionally preceded by less defined unfolding events (arrowhead). dtc., detachment peak. **b** Representative F-D spectra obtained in the absence of Ca²⁺. Schematic representations of hypothesized interactions between the transmembrane helices are shown below. Dots indicate the approximate location of the force peaks that do or do not undergo major changes upon Ca²⁺ binding (dashed and solid symbols,

respectively). **c, d** As in a but in the presence of 2 mM Ca²⁺ (*n* = 70). **e** Time-lapse analysis of TMEM16F subunits height (S₁ and S₂ in green and gray colors, respectively) and changes in lipid volume (ΔVL, black color). Upon Ca²⁺ injection, subunit S₂ moves away from the membrane plane by ~1 nm and concurrently remodeling of lipid bilayer (LR) in and around the TMEM16F dimer is observed (black symbols in plot e and dotted white outline in insets ii-iv). These morphological changes in the membrane around TMEM16F dimers are possibly related to TMEM16F lipid scrambling and were observed in two independent experiments. In agreement with the large variability in functional measurements reported in Fig. 5a, b such effect was not observed in all molecules/experiments. **f** Height distributions for S₁ (left panel) and S₂ (right panel) suggesting independent subunits motion and two main states at around ~2.8 and ~3.7 nm height. Sky blue and salmon colors refer to distributions observed in the presence of EGTA and Ca²⁺, respectively.

TMEM16F molecules. Indeed, imaging of the channel from the intracellular side is expected to detect conformational changes occurring on the surface of the large cytosolic NCD, whose unfolding was less reproducible and therefore uninformative. TMEM16F was first identified and imaged at high resolution in the absence of Ca²⁺ (Fig. 4e and

inset (i)). The two subunits (S1 and S2 in Fig. 4e) and the characteristic membrane depression in-between can be readily identified. Remarkably, the dimer adopts a loose, o/r configuration (see also Fig. 3) and features an asymmetric subunit state: the cytoplasmic domain of S1 protrudes ~3.7 nm from the membrane plane whereas S2 does ~2.8 nm

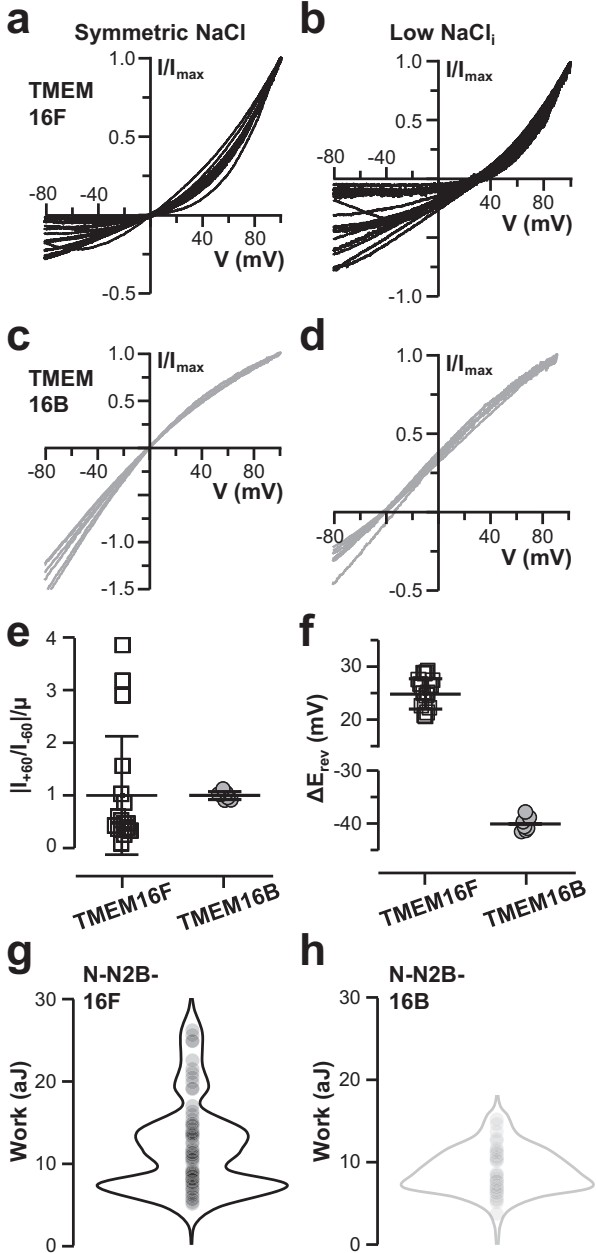

**Fig. 5 | The variability of electrical properties in TMEM16 channels. a, b** Inside-out excised membrane patches from transiently transfected HEK293 cells expressing TMEM16F were recorded in symmetrical NaCl (140 mM, **a**) and lower intracellular NaCl (14 mM, **b**) solutions. Their IV relations were determined via voltage ramps from −80 mV to +100 mV. Currents were activated by 1 mM $CaCl_2$ and normalized to the current at +100 mV. **c, d** The same as **a, b** but for TMEM16B. **e** Comparison of the changes of rectification for TMEM16F (black) and TMEM16B (gray) currents. Rectification was calculated as the ratio between currents measured at +60 and −60 mV ($|I_{+60}/I_{-60}|$) and normalized to the average rectification value μ (the average current at −60 mV is −266 ± 199 pA for TMEM16F, −430 ± 286 pA for TMEM16B and −0.2 ± 2 pA for not transfected cells (nt); at +60 mV is 673 ± 455 pA for TMEM16F, 293 ± 221 pA for TMEM16B and −0.6 ± 3 pA for nt cells, $n = 18$ for 16F, $n = 7$ for 16B an $n = 6$ for nt). **f** Comparison of the shift of reversal potentials after the replacement of 140 mM NaCl with 14 mM NaCl for TMEM16F and TMEM16B ($n = 19$ for 16F and $n = 7$ for 16B). In panels 5e,f mean values +/- standard deviation are overlaid to individual data points. **g, h** Violin plots of unfolding work of $Ca^{2+}$-bound N-N2B-16F ($n = 70$, average W 12.0 aJ, **g**) and N-N2B-16B ($n = 38$, average W 8.8 aJ, **h**).

(Fig. 4f, sky blue histograms). Although TMEM16 cryo-EM structures usually present a symmetrical arrangement, these results are in line with most recent structural investigations evidencing asymmetric subunit configurations[29,30].

After continuously imaging the TMEM16F intracellular face for ~2 min without observing any change, we added $Ca^{2+}$ to a final concentration of ~2 mM (Supplementary Movie 2; Fig. 4e). Shortly after the addition (t ~ 60 s, Fig. 4e), a change in the surface topography emerges; subunit S2 undergoes a ~1 nm vertical excursion away from the membrane plane, matching subunit S1 height of ~3.7 nm, whereas the latter remains unchanged (Fig. 4e, compare green and gray traces). Height's fluctuation histograms (Fig. 4f) indicate two dynamic states of TMEM16F protomers (i.e. ~2.8 and 3.7 nm, Fig. 4f), and lend further support to previous studies suggesting that both TMEM16 subunits can be independently activated by $Ca^{2+}$ (refs. 67,68). Further inspection of surface topographies reveals a marked restructuring of the membrane bilayer morphology around the TMEM16F dimer (Fig. 4e, insets ii-iv). Upon $Ca^{2+}$ supplementation, substantial lipid remodeling evolved in concert with the observed vertical motion of subunit S2 (Fig. 4e, black lines and symbols). We argue that the change in membrane structure -which phenomenologically appears as a membrane lifting of about 0.6–0.8 nm confined around the TMEM16F dimer - might be related to a lipid phase transition and/or membrane-driven blebbing resulting from lipid scrambling activity[8,69].

Collectively, our results suggest that the TMEM16F NCD structure is flexible and dynamic. They present visual evidence of scramblase activity at the single-molecule level in TMEM16F dimers characterized by a loose, o/r subunit arrangement. Finally, SMFS experiments suggest $Ca^{2+}$-induced rearrangements of TM3, TM4 and TM6 helices, extending previous structural results to native membranes and physiological-like conditions.

## Electrical and mechanical properties of TMEM16F are more variable than TMEM16B

To test whether the structural heterogeneity of TMEM16F suggested by our SMFS and HS-AFM data was related to ion permeation, we performed electrical measurements from TMEM16F and TMEM16B. The latter also expresses in NG108-15 cell line (Supplementary Fig. S2a and S3b), but unlike TMEM16F, TMEM16B is scrambling incompetent and only functions as an ion channel[25,70].

Electrical recordings were performed on transiently transfected HEK293 cells about 48 h after transfection. Ionic currents and permeation properties were evaluated in excised membrane patches by voltage ramp experiments under saturating $Ca^{2+}$ concentration (Fig. 5a–d). In symmetrical NaCl, currents from TMEM16F exhibited a varying degree of rectification (Fig. 5a), which was less prominent in currents obtained from TMEM16B channels (Fig. 5c). Specifically, the current-voltage (I-V) relationship of TMEM16F displayed varying degrees of outward rectification up to a strong voltage-dependency with minimal current flow at negative membrane potentials (Fig. 5a). In stark contrast, TMEM16B exhibited currents with highly reproducible inward rectification (Fig. 5c). The difference in rectification variability between TMEM16F and TMEM16B channels is better seen when normalized rectification indexes ($|I_{+60}/I_{-60}|/\mu$) of each replicate were compared (Fig. 5e). Indeed, data dispersion, as indicated by standard deviations, showed large differences between TMEM16F and TMEM16B channels, being equal to 1.12 and 0.08, respectively (Fig. 5e). Furthermore, a large variability in TMEM16F was also observed when examining the reversal potential shifts ($\Delta E_{rev}$) after replacing the intracellular bathing solution from high (140 mM) to low (14 mM) NaCl (Fig. 5b, f). As shown in previous studies, being TMEM16B overall more selective for $Cl^-$ and TMEM16F for $Na^+$ (refs. 71,72) a significant difference in $\Delta E_{rev}$ average values was observed (Fig. 5b, d, f). However,

whilst $\Delta E_{rev}$ were highly consistent among the different trials in TMEM16B (Fig. 5d), with a relatively small coefficient of variation ($\sigma^2/\mu$) value of 0.03, $\Delta E_{rev}$ for TMEM16F exhibited substantial variability which translates into the much larger $\sigma^2/\mu$ value of 0.112 (Fig. 5f). Similar results were obtained when TMEM16F was activated by sub-saturating $Ca^{2+}$ concentration, suggesting that TMEM16F may have similar structural flexibility at both low and high $Ca^{2+}$ (Supplementary Fig. S10).

Having found these differences in electrical properties, we decided to investigate the unfolding work of the scrambling-incompetent TMEM16B channel. To this end, we conjugated the N2B fingerprint to TMEM16B gene (His$_6$-N2B-TMEM16B-GFP), overexpressed this construct in NG108-15 cells (Supplementary Fig. S11a), and performed unfolding experiments in saturating concentrations of $Ca^{2+}$ (Supplementary Fig. S11b). We found that despite a similar 3D architecture and a 33% sequence identity (Supplementary Fig. S12), TMEM16B and TMEM16F have different unfolding behaviors. Indeed, TMEM16F F-D curves had an average unfolding W of 12.0 aJ and a relatively broad bimodal distribution - in the range of 4-28 aJ (Fig. 5e) - whereas TMEM16B curves exhibited an average W of 8.8 aJ and much narrower dispersion, in the range of 3-17 aJ (Fig. 5h). Thus, with an almost 1.7 times broader distribution, TMEM16F unfolding appears much more variable compared to TMEM16B.

Altogether, these data establish a direct connection between the observed structural heterogeneity and TMEM16F function. Indeed, if we were to simplistically assume that the two dominant unfolding W subpopulations of TMEM16F differ in their degree of voltage-dependency (or permeability), depending on the specific mix of the two channel subsets present in the inside-out membrane patch, the degree of the measured current rectification (or permeability ratios) would vary from experiment to experiment. Thus, the large variability in ion permeation properties as compared to those of the TMEM16B channel, potentially originates from TMEM16F large structural flexibility, captured by both HS-AFM imaging and SMFS unfolding experiments.

## Discussion

The present study addresses the conformational dynamics of TMEM16F scramblase under physiologically relevant conditions. To achieve this objective, we build upon membrane isolation and in situ protein identification methods established by our group[56] and took advantage of the capabilities of AFM, enabling direct observation and manipulation of biomolecules at single-molecule level. This allowed us to unveil variability in the quaternary structure and the conformational dynamics of TMEM16F, pinpoint its unfolding barriers and how they change upon ligand binding, with a resolution down to a few amino acids. Despite inherent limitations, including the restriction of AFM imaging to surface structures and uncertainties related to protein-probe interactions within complex native cellular membranes, we provide several lines of evidence that TMEM16F samples a broad structural ensemble, some conformations of which greatly differ from previously determined cryo-EM structures. The observation of substantial lipid remodeling and deformation around these "unorthodox" dimeric assemblies and the measured variable electrical properties in response to $Ca^{2+}$ binding, further indicates that such structural plasticity is genuine and important for TMEM16F catalytic activity. Therefore, it is likely that this conformational heterogeneity − which is possibly a common physical feature embodied by other TMEM16 family members − might be linked to TMEM16F's dual channeling and scrambling function.

Assignment of the unfolded segments to TMEM16F secondary structure suggests rather drastic movements of TM3, TM4, and TM6 helices upon $Ca^{2+}$ binding. These transitions are reminiscent of those observed in TMEM16 fungal scramblases, where separation of TM4 and TM6 in the $Ca^{2+}$-bound state leads to opening and exposure of the subunit cavity to lipid headgroups[20,22]. Specifically, in the presence of $Ca^{2+}$ our pulling experiments revealed an unfolding event at 189 nm with an increased frequency from 0.33 in TMEM16F to 0.75 in the Y563K gain-of-function mutant. This unfolding peak locates into the extracellular end of TM3 and allows us to draw two conclusions: (i) in high $Ca^{2+}$, TMEM16F populates two major conformations (i.e. with or without the 189 nm peak) that differ from the $Ca^{2+}$-free state in the pore region (see Fig. 4); (ii) the higher occurrence of the 189 nm force peak in the activating mutant indicates that further repositioning of TM3 and TM4 is likely key to support phospholipid scrambling. The latter finding is well in agreement with recent structures of some TMEM16F gain of function mutants, which evidenced kinking of the extracellular part of TM3 - around G473 - and straightening of TM4 in response to $Ca^{2+}$ binding[30]. Inspired by structural studies on fungal scramblase and atomistic in silico investigations on TMEM16F (refs. [73,74]), we propose that these mechanical changes underscore a stepwise disengagement of TM4 from TM6 to dilate and possibly fully open-up the permeation pathway to the membrane. Accordingly, phospholipid transbilayer transfer across the open cavity furrow would proceed through a canonical in-groove mechanism[3,15,25]. Although these findings conform well to the alternating pore/cavity model where closed, intermediate ion-conductive, and fully open lipid-conductive states exist in equilibrium (Fig. 1b), further experiments will be needed to resolve whether or not the intermediate state here observed can support ion permeation and if the mechanical changes of TM3 and TM4 segments lead to a fully or only partially open, lipid-exposed conduit[30,73,74].

Yet, our results differ from the alternating pore/cavity model and any other mechanism based on structural studies recently proposed, to the extent that HS-AFM and SMFS identified dynamic changes of the dimerization interface, protomers orientation, and quaternary assembly of TMEM16F that were not previously anticipated. We speculate that experimental cryo-EM conditions might have stirred the TMEM16F energy landscape toward the population of more "compact" assemblies, thereby overlooking other important structural subsets. Additionally, current multi-conformation reconstruction algorithms based on clustering approaches to few discrete structural classes might be inadequate to deal with specimens exhibiting extensive or quasi-continuum structural flexibility. Deficiencies of current cryo-EM methods to capture alternative conformations predicted by electrophysiology and other techniques has been recently ascertained for ionotropic glutamate receptors[75].

Nonetheless, our findings do not seem to be refuted by current structural data. Comparison of X-ray and cryo-EM structures of TMEM16K scramblase shows significant reorientation of the protomers composing the dimeric assembly, including a ~10° rotation of the NCDs, as well as changes in the oligomerization interface[18]. These transitions were linked to opening of the subunit cavity and, although they refer to a different TMEM16 homolog, are in qualitative agreement with our results. On a related note, inspection of TMEM16F structures reveals that the oligomerization interface is fairly small and limited to the extracellular end of TM10, with the crevices in between filled with lipids[14]. This observation is fully compatible with our findings as it suggests that the molecular interactions stabilizing the cryo-EM homodimers are rather weak. Additionally, the structural plasticity of NCDs might further contribute to the different quaternary arrangements identified by our SMFS and HS-AFM analysis. Indeed, high-resolution structures of TMEM16F feature several poorly defined loops connecting short secondary elements within the NCD, which is suggestive of a highly dynamic domain (Supplementary Fig. S12). Consistently, we found that its unfolding occurs at forces below 20 pN and does not show well-defined and reproducible force peaks (see Fig. 4a−d), which is indicative of a mechanically unstable region. Physical considerations and molecular dynamic simulation suggest that in aqueous solutions such soluble domains experience significant stochastic fluctuations and conformational entropy[76]. Based on this

evidence, we argue that the structural flexibility of the NCD endows the multiplicity of intersubunit interactions and dimerization interfaces here reported.

Having established that large-scale protomer motions and changes at the dimerization interface occur and are likely, an obvious question arises: what would be their functional significance? A recent computational study suggests that binding of a third $Ca^{2+}$ on TM2 and TM10 in TMEM16F – i.e. at the cryo-EM dimer interface (Fig. S1) – allosterically inhibits the opening of the subunit cavity[73], providing a possible explanation. When the two primary $Ca^{2+}$ binding sites on TM6-TM8 remain occupied, the opening of the occluded permeation pathway will be energetically facilitated by and coupled to the thermal disengagement of the third $Ca^{2+}$ ion from this regulatory site. Within this scenario, subunits reorientation would directly disrupt this regulatory $Ca^{2+}$ binding site, favouring the widening and exposure of the subunit cavity to the lipid bilayer, ensuing the efficient translocation of phospholipids. Additionally, our structural assembly models (Fig. 3) offer a more speculative and tantalizing hypothesis. Specifically, we postulate that the observed clamshell opening of the dimer might unleash an additional scrambling interface between the two TMEM16F subunits. This proposal is supported by the observation of straddled lipids wedged within the dimer interface cavity in several TMEM16 structures[14,16,32]. Because of their nearly perpendicular orientation, usually scrambling is not though to occur through this interface. However, upon protomers rotation and the consequent subunits disengagement some sort of bilayer rearrangement is likely. Indeed, AFM imaging revealed a large depression between the two TMEM16F subunits in the loose dimer configuration, compatible with a ~1 nm membrane thinning (Figs. 2, 4). This bilayer distortion could potentially facilitate phospholipids transbilayer movement through an additional out-of-the-groove scrambling mechanism, explaining how bulky PEGylated lipids too large to fit within subunit cavity are translocated[31].

In sum, our results suggest that TMEM16F has a shallow energy landscape, leading to a hitherto overlooked conformational heterogeneity where different structural subpopulations exist in a dynamic equilibrium. This scenario has been previously invoked to conceptualize the structural multiplicity, signaling richness and complex pharmacology of human β2-adrenergic receptor[77], and in the case of TMEM16F, might provide a simple yet elegant explanation to seemingly contradictory results reported in functional assays. Indeed, many biophysical properties, such as ionic selectivity, current rectification and kinetics, and $Ca^{2+}$ sensitivity have been shown to greatly depend on the specific experimental conditions employed[72,78]. We propose that the structural plasticity of TMEM16F underscores its functional promiscuity. Local cellular regulators, such as $PIP_2$ (ref. 79) and other yet-to-be-identified drivers, might operate by conformational selection and fine-tune TMEM16F function to fill specific tissues and cellular subpopulation requirements. Therefore, structural multiplicity establishes the physical foundation for a tight spatial and temporal control of TMEM16F's various physiological functions.

## Methods

### Molecular biology and TMEM16 constructs
cDNAs were cloned into the expression vectors peGFP-N1, peGFP-C1, and pCMV-Sport6 (Supplementary Table 2) according to our research goals. The Mus musculus TMEM16F (Ano6, NP_780553.2) and TMEM16B (Ano2, NP_705817.1) consists of 911 a.a. and 913 a.a., respectively. The DNA plasmids were constructed as described previously[47]. A His6-N2B tag (210 a.a.) that is composed of six histidines (His6) and an N2B module of giant muscle protein titin is added at the N- or C-terminus of the cDNA according to our research purpose. These TMEM16 constructs with His6-N2B tag were made and their sequences were confirmed by Genewiz Company in Suzhou, China. The His6-N2B tag doesn't display any unfolding events under AFM stretching as described before[47,58].

### Cell culture and transfection
Mouse and rat hybrid neuroblastoma NG108-15 cells (Sigma-Aldrich) and HEK293 cells (American Type Culture Collection) were cultured in medium composed of Dulbecco's Modified Eagle Medium (DMEM, Gibco), GlutaMax-l (ThermoFisher), 10% fetal bovine serum (FBS, Gibco), 100 U/ml Streptomycin and 100 U/ml Penicillin. The DNA constructs of interest (Supplementary Table 2) were transiently transfected into the cells grown on coverslips by using Lipo2000 or X-tremeGENE (Roche). For pCMV-TMEM16B-Sport6 transfection, HEK293 cells were co-transfected with peGFP-N1 for fluorescent identification of transfected cells[72]. Regarding TMEM16F Y563K mutant, after 6 h from the transfection the old culture medium was replaced by $Ca^{2+}$-free medium to avoid cytotoxicity[65]. Cells were cultured into a humidified incubator (5% $CO_2$, 37 °C) and after 48 h-72 h transfection, subjected to AFM-based SMFS, immunofluorescence, or electrophysiological recording. The cell line used in this study (NG108-15, Cat# 88112302; HEK-293, Cat# 85120602) were purchased from European Collection of Authenticated Cell Cultures (ECACC) through its distributor Sigma Aldrich and were authenticated by the supplier using standard authentication methos, such as short tandem repeat (STR) profiling.

### Immunofluorescence
The immunostaining was carried out following previously reported procedures[80]. The NG108-15 cells grown on coverslips were fixed by 4% PFA, permeabilized with 0.05% TritonX-100 and then blocked by blocking buffer (10% FBS + 5% BSA) for 90 min at RT. The cells were incubated with specific primary antibodies at a ratio of 1: 400 (TMEM16B Cat# 20647-1-AP, Proteintech Euro; TMEM16F Cat# ACL-016, Alomone Labs) or 1x PBS (Control) at 4 °C overnight, followed by rinsing with pre-cold 1xPBS 3 times. Fluorescent staining was developed with the secondary antibody Alexa 594-labeled goat anti-rabbit (1:800, Cat# A11037, Invitrogen), Abberior STARRED (1:200, Cat# STRED-1002-20UG, Abberior GmbH), Abberior STARGREEN (1:200, Cat# STGREEN-1002-20UG, Abberior GmbH) at RT for 90 min, as indicated. Membrane and cell nuclei were labeled with STARRED membrane (1:200, Cat# STRED-0206-100PMOL, Abberior GmbH), and Hoechst (1:400, Cat# 33342, Thermo Scientific) or DAPI (1:1000, Cat# 32670, Sigma), respectively. Cells and isolated membrane fragments were imaged with a Nikon A1R microscope with 60x oil immersion (NA1.40) and 20x objectives, respectively. Stimulated emission depletion (STED) imaging and confocal z-sectioning was performed with a STEDYCON microscope (Abberior GmbH). Images were acquired with NIS-Elements (Nikon) and STEDYCON (Abberior GmbH) acquisition software and analyzed with ImageJ 1.47 v (NIH) and Image Gallery 9 (Abberior GmbH).

### Protein reconstitution
Mouse TMEM16F reconstituted in liposomes were prepared according to previous protocols[14,30] and obtained from Prof. Raimond Dutzler (University of Zurich). Briefly, dry lipid films (4:1 mol/mol mix of soybean polar lipid extract: cholesterol) were solubilized in 20 mM HEPES pH 7.4, 300 mM KCl, and 2 mM EGTA (Buffer A) at 20 mg/ml by sonication and three freeze-thaw cycles. The lipid suspension was extruded through a 100 nm pore polycarbonate membrane and diluted to 4 mg/ml in Buffer A. Liposomes were destabilized in a spectrofluorometer by titration with Triton X-100 until decrease scattering at 540 nm was observed. Subsequently, 0.08% Triton X-100 and detergent-solubilized TMEM16F were added to a w/w lipid-to-protein ratio (LPR) of 100:1 or 13:1. According to Western blot analysis, the lowest LPR achieved was ~ 50:1. Detergent removal was performed by adsorption onto biobeads SM2 (Bio-Rad). After 24 h, the suspension was filtered and proteoliposomes were harvested by centrifugation at 160,000 g for 30 min. The pellet was resuspended in Buffer A to a final lipid concentration of 10 mg/ml and stored at −80 °C until further AFM measurements.

## SMFS unfolding experiments and analysis

The apical membranes of NG108-15 cells were isolated with the unroofing method described in refs. 55,56. Briefly, glass coverslips (24 mm in diameter, 170 μm in thickness) were split by hand into a triangular disk sector (in this way the edges are optically sharp). The triangular coverslips thus obtained were mounted on the AFM (JPK Nanowizard 3) head and used to squeeze the target cell. Apical cell membranes were isolated by abrupt withdrawal and the coverslip with the membrane fragments was laid down and fixed on the Petri dish. The medium was replaced in $Ca^{2+}$ AFM unfolding buffer (145 mM NaCl, 3 mM KCl, 2 mM $CaCl_2$, 10 mM HEPES, pH 7.4) or $Ca^{2+}$-free AFM unfolding buffer (145 mM NaCl, 3 mM KCl, 1 mM EDTA, 10 mM HEPES, pH 7.4).

The AFM experiments were performed by using gold-coated AFM cantilevers (APPNano, Cat#HYDRA2R-50NGG, spring constant ~0.08 N/m). After calibrating the inverse optical lever sensitivity of the AFM by doing a force curve on the glass surface, and the lever with the commonly used thermal noise-based method in working medium[56], the sharp tip was used to detect the membrane in a non-contact mode (~15 KHz). SMFS experiments were performed at ~24 °C in AFM unfolding buffer with or without $CaCl_2$, as indicated. Pulling velocity of 0.6 μm, indentation force of ~1 nN, and extend delay time of 0.6 s to favor protein physisorption[56] were used. All SMFS data were initially collected with JPKSPM control software (JPK-Bruker) and then imported and processed with the software Fodis[81] in Matlab 2017b (Math-Works). Traces obtained from transfected cells with the N2B-TMEM16 construct (n. Traces = 890603 for $Ca^{2+}$-free and 922109 for $Ca^{2+}$-bound TMEM16F, 441015 for $Ca^{2+}$-bound TMEM16B) were filtered to identify the N2B tag (force lower than 40 pN between 20 and 70 nm) in traces longer than 340 nm (these traces are shown in Fig. 2b). We only performed SMFS on the flattest areas of the membrane patches to limit uncertainties arising from membrane corrugations and height variation (Fig. 1c) on this filtering step. After verifying that in the control non-transfected cells (n. Traces = 846997 for $Ca^{2+}$-free and 780059 for $Ca^{2+}$-bound TMEM16F) the same filtering procedure did not return any trace, an automatic clustering[57] and Bayesian identification procedure[56] were applied in order to find the native cluster of TMEM16F which matched the unfolding pattern found in the transfected cells (see Fig. 1i) shifted by the length of the N2B (cluster 8, $n = 85$; default parameters can be found in the header of https://github.com/ninailieva/SMFS_clustering/blob/master/cluster_traces.cpp). The clustering algorithm used consists of five major blocks. In the first block, the initial negative-force parts of the F-D curves not related to the unfolding process are removed, and a coarse filtering aimed at the detection of spurious F-D curves is performed (curves with tilted baselines, with oscillations due to environmental noise, etc.). In the second block, a quality score based on the agreement of the experimental data with the Worm Like Chain model is computed and assigned to each trace. This score is used to select physically meaningful traces for further analysis. In the third block, distances between pairs of traces are computed to assess their similarity. These similarity distances are used in the fourth block for density peak clustering which is a technique to find groups of similar F-D curves in an unsupervised manner. The fifth and final block consists in the refinement and possibly in the merging of some of these clusters that are indistinguishable by human inspection. In purified and reconstituted TMEM16F (n. traces = 430181) we identified only one main unfolding pattern that correspond to the pattern found in WT NG108-15 cells (see Supplementary Fig. S5) using the fingerprint roi function in Fodis which classify traces by similarity setting a template. No sawtooth-like patterns were observed in retraction curves from Soybean supported bilayers obtained from empty liposomes devoid of mTMEM16F (n. traces = 27332). The Unfolding work (W) was calculated in Fodis environment as the integral of the F-D curves from 0 nm up to 400 nm (after cantilever detachment). The contour length of the unfolded segments −

peaks in the FD traces − were identified and collected in global histograms of contour length according to previous methods[47]. The resulting global histograms were fitted with a Gaussian mixture model, and peaks occurrence within one standard deviation from the center of each Gaussian bell are reported as probabilities for the unfolding intermediates. Topographic images of isolated membrane fragments were processed and analyzed with ImageJ 1.47 v (NIH) and Gwyddion 2.58 (64 bit).

This method is inherently incompatible with unfolding proteins with both C- and N-termini on the extracellular side of the membrane. One solution could be to perform an AFM tip approach on a living cell at nN force range. However, during retraction, the upper cell membrane is pushed down a few hundred nm and vice versa. The unfolding spectra of a membrane protein will therefore be superimposed by an unpredictable change in offset due to the movements of the upper cell membrane, making accurate data interpretation difficult.

## HS-AFM imaging and image processing

All AFM observations were performed in tapping mode using a laboratory-built apparatus previously described[82]. Briefly, a cylindrical glass sample stage (diameter, 2 mm; height, 2 mm) with a thin mica disc (diameter, 2 mm; thickness ~ 0.1 mm) fixed on top was attached onto the upper face of the z-scanner by a drop of nail polish. A 3 μl drop of diluted proteoliposomes (1 mg/ml) was deposited onto the mica surface (freshly cleaved) and incubated for 20 min. After adsorption the sample was thoroughly rinsed with $Ca^{2+}$ AFM imaging buffer (150 mM NaCl, 2 mM $CaCl_2$, 20 mM HEPES, pH7.4) or $Ca^{2+}$-free AFM imaging buffer (150 mM NaCl, 1 mM EDTA, 20 mM HEPES, pH7.4) to remove excess lipids. The stage with the TMEM16F-containing planar bilayer on top was then immersed in a liquid cell filled with ~100 μl imaging AFM buffer which was or not supplemented with $CaCl_2$ (as indicated). Short cantilevers (BL-AC10DS-A2, Olympus) with nominal spring constant of ~100 pN/nm, resonance frequency of ~0.5 MHz, and quality factor of ~1.5 in water were used[82]. An amorphous carbon tip was fabricated on the original AFM tip by electron beam deposition (~500 nm in length and tip radius of ~4 nm) and etched under argon plasma (Tergeo, PIE Scientific) to further sharpen the apex down to ~1.5−2 nm radius. The cantilever's free oscillation amplitude $A_0$ and set point amplitude $A_s$ were set at ~2 nm and around $0.9 \times A_0$, respectively. Under these conditions the energy delivered by a tip-sample interaction is 1−3 $k_BT$ on average[82].

Data were collected using laboratory-developed software based on Igor Pro 8 (WaveMetrics) and Visual Basic.NET (Microsoft)[82,83]. HS-AFM movies were x,y-drift-corrected using the ImageJ plugins "Template Matching and Slice Alignment" (https://sites.google.com/site/qingzongtseng/template-matching-ij-plugin). Image flattening was achieved by means of plane or second order polynomial surface fitting (as appropriate) followed by median (0 order) line-by-line levelling to remove remaining height offsets along the fast scan axis[84]. These steps were performed using in-house software routines developed in MATLAB R2017b and R2022a (MathWorks), available at https://github.com/arin83/U1067/. Changes in lipid volume and TMEM16F subunits heights (Fig. 4e, f) were calculated using the standard measurement tools and built-in functions in ImageJ 1.52e (NIH) and Matlab R2022a (MathWorks).

## Structural models of TMEM16F dimers

To infer atomistic structures from HS-AFM images simulation AFM and automatized fitting within the BioAFMviewer software[63,64] was employed using the cryo-EM data PDB ID 6p46 as a template. Simulated scanning was based on non-elastic collisions of a rigid cone-shaped tip (cone-half angle 5°, probe sphere radius 1 nm) with the rigid Van-der-Waals atomic model of the protein structure. Automatized fitting was based on identifying the best match of the simulated image with the target HS-AFM image, quantified by the image correlation

coefficient. Thus, the atomic structure behind the experimental topography was obtained. For fitting into the cryoEM-like (em) HS-AFM topography, the complete dimeric channel structure of 6p46 was used. For the HS-AFM images corresponding to the slid interface (si) and open/rotated (o/r) conformations, the two chains were separately fitted into the topographies to obtain structural models with altered domain arrangements. From individual fits the combined structural model was reconstructed from the known shift in translation and relative orientation of the two chains. Cryo-EM structures shown in Supplementary Fig. S1 were visualized and rendered with PyMOL 2.5.4 (Schrödinger).

### Electrophysiology

Electrophysiological recordings were performed on excised inside-out patches from HEK-293 cells transfected with TMEM16F or TMEM16B with or without His6-N2B tag. Patch pipettes were made of borosilicate glass (WPI, Sarasota, FL, USA) with a PP-830 puller (Narishige, Tokyo, Japan). Currents were recorded with an Axopatch 1D and controlled by Clampex 9 via a Digidata 1332 A (Axon Instruments, Union City, CA, USA). Data were low-pass-filtered at 4 kHz and sampled at 10 kHz. Experiments were performed at room temperature (20–22 °C). The bath was grounded via a 3-M KCl agar bridge connected to a Ag/AgCl reference electrode. The cells were continuously perfused with mammalian Ringer's solution containing (in mM): 140 NaCl, 5 KCl, 2 CaCl$_2$, 1 MgCl$_2$, 10 Hepes, and 10 Glucose pH 7.4. The patch pipette contained (in mM): 140 NaCl, 5 EGTA, and 10 Hepes, pH 7.2. The bathing solution at the intracellular side of the patch contained (in mM): 140 or 14 NaCl, 10 HEDTA, and 10 Hepes, pH 7.2, and no added Ca$^{2+}$ for the nominally 0 Ca$^{2+}$ solution, or 1 mM CaCl$_2$. In low NaCl solution the osmolarity has been maintained by adding sucrose. Liquid junction potentials were calculated using the pCLAMP 9 software (Axon Instruments, Union City, CA, USA), and applied voltages were corrected off-line. Changes between different solutions were performed using the Perfusion Fast-Step SF-77B (Warner Instrument Corp., Holliston, MA, USA). For voltage ramp IV relations, we exposed the patches to Ca$^{2+}$-containing solution for 1 s at +100 mV and then applied a ramp from +80 to -80 mV at 0.36 mV/ms. Leak currents measured in nominally 0 Ca$^{2+}$ solution were subtracted. To reduce the possible error due to the current rundown, we performed the experiments after the fast component of run-down (about 1 min) when the current reach reasonable steady-state. Data were analyzed with Microsoft Excel 365, and IgorPro 6.3 and 8 (WaveMetrics) software.

### Reporting summary

Further information on research design is available in the Nature Portfolio Reporting Summary linked to this article.

## Data availability

The manuscript figures, supplementary information, and source data files contain all data necessary to interpret, verify, and extend the presented work. The raw AFM data is saved as .jpk-force and .asd files and therefore can only be opened using proprietary software. These data are available from n.galvanetto@bioc.uzh.ch (SMFS dataset) and a.marchesi@staff.univpm.it (HS-AFM data) upon request. The source data underlying Figs. 1c, e–i, 2b, e, 3a–c, 4a–f, 5a–h, and Supplementary Figs. S1c, S2e, S6b, S7a–d, S8b, S9b, S10a–d are provided as a Source Data file. Published experimental structures of TMEM16F identified by the following PDB codes were used in this study: 6P46, 6QP6, 6QPB Source data are provided with this paper.

## Code availability

Data analysis was performed with customs scripts in Matlab R2017b and R2022a. All codes are published[57,81,82,84] and available on public repositories (https://github.com/arin83/U1067/; https://github.com/galvanetto/Fodis; https://github.com/ninailieva/SMFS_clustering).

The HS-AFM acquisition software[82,83] is available from the corresponding author (a.marchesi@staff.univpm.it) upon request.

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

## Acknowledgements

We sincerely thank Prof. Raimund Dutzler (University of Zurich) for providing TMEM16F proteoliposomes. We are also most thankful to Prof. Huanghe Yang (Duke University) for sharing the TMEM16F mutant and Prof. Guidalberto Manfioletti (University of Trieste) for providing the peGFP-N1 plasmid. The authors are also deeply indebted to Professors Toshio Ando and Mikihiro Shibata at Kanazawa University for providing access to HS-AFM apparatus and Abberior GmbH (Gottingen) for supplying antibodies and the loaner of Stedycon STED microscope. We extend our gratitude to Frédéric Eghiaian (Abberior GmbH) for invaluable guidance with confocal/STED image acquisition and treatment. L.P., H.F., C.M.F., and A.Marchesi acknowledge financial support by the Japanese Ministry of Education, Culture, Sports, Science and Technology (World Premier International Research Center Initiative WPI). S.G. is supported in part by the National Natural Science Foundation of China (52071332) and in part by the Department of Science and Technology of Guangdong Province (Grant Nos. 2019QN01H430). H.V. acknowledges the following funding: the National Natural Science Foundation of China and Swiss National Science Foundation (NSFC-SNF 32161133022); the Shenzhen Key Laboratory of Computer-Aided Drug Discovery, Advanced Technology, Chinese Academy of Sciences, Shenzhen (funding no. ZDSYS20201230165400001); the Chinese Academy of Science President's International Fellowship Initiative (PIFI no. 2020FSB0003); Guangdong Retired Expert (granted by Guangdong Province); Shenzhen Pengcheng Scientist; the AlphaMol and SIAT Joint Laboratory; the Shenzhen Government Top-Talent Working Funding and Guangdong Province Academician Work Funding.

## Author contributions

Z.Y. performed SMFS experiments; N.G. developed techniques and analysis tools to perform SMFS experiments in native cell membranes; A. Marchesi and L.P. performed HS-AFM imaging experiments; M.A. purified and reconstituted protein; C.A.S.T. and S.P. performed electrophysiology; M.D.P., C.A.S.T., and Z.Y. performed immunofluorescence; H.F. performed modeling and structural analysis; C.M.F., N.G., A.Marchesi, A.Menini, L.P., S.P, V.T., Z. Y. analyzed the data; C.M.F., L.P., H.V., S.G., and A.Menini edited the manuscript; Z.Y., N.G., V.T and A.Marchesi designed the research; V.T. supervised the project; Z.Y., V.T and A.Marchesi wrote the manuscript with input from all authors.

## Competing interests

The authors declare no competing interests.
