## [Peer Review File · Nature Communications]

Structural heterogeneity of the ion and lipid channel
TMEM16FReviewers' Comments:

Reviewer #1:

Remarks to the Author:

Overall assessment:

Ye, et al offer a unique and intriguing addition to the TMEM16F structure/function literature, providing alternative evidence to attempt to bridge seemingly incompatible findings concerning TMEM16F substrate permeation and gating. Using primarily AFM, the authors find TMEM16F occupies a broader conformational landscape than those previously identified in cryo-EM structures. In particular, they find variability in the dimerization interface between monomers, where they show "slid" and "open/rotated" configurations using high speed AFM imaging. Their data support previous findings of subunit asymmetry in the dimer and that monomers function independently from one another. While the experimental approach is unique and informative, the following issues need to be further addressed or clarified.

Major comments:

1. Title: TMEM16F is a scramblase and channel. The authors discussed both channel and scramblase activities. Just emphasizing channel in the title seems to be biased.
2. Cell line chosen: The authors used NG108-15 cell line, which highly expresses endogenous TMEM16B and TMEM16F (Fig. S2a). It was not justified why chose this cell line to overexpress the N2B constructs instead of using a cell line deficient of TMEM16F and B? Will the heterologously expressed proteins coassemble with the endogenous TMEM16 proteins to form hetero-dimer (one endogenous subunit, one N2B subunit)? Is it possible that the expressed TMEM16F form heterodimer with endogenous TMEM16B? What is the evidence that the heterodimerization does not exist? If it indeed exists, will it complicate the interpretation of the AFM results?
3. Sample preparation: In the unfolding experiments, how tightly the extracellular loops of TMEM16F is anchored to the coverslips? The authors mentioned that the maximal contour length of ~370-390 nm (Fig. S3) is roughly corresponding to the unfolding of the whole TMEM16F (Fig. 1e). However, if the membrane and protein partially detach from the coverglass, the F-D is no longer reliable. Please clarify/explain.
4. Please also introduce what are the rational and evidence that a membrane protein embedded in the membrane has to unfold stepwise according to the order of the sequence. This is critical for assigning the F-D peaks to different TMs and interpreting the conformational changes of TM3, 4, 6 in Figure 4.
5. Why there are only a few unfolding events recorded, given the large folded the structure at the cytosolic domain? Does this imply the cytosolic N-terminus cannot unfold? TMEM16s have 10 TMs, why usually only 5-6 peaks were observed in the F-D curves?
6. Page 8, The authors wrote "the unfolding process and the occurrence of force peaks are stochastic in nature". However, in Page 6, the authors described "a reproducible sequence of force peaks was observed for both, WT-16F and N-N2B-16F with peaks varying much in amplitude but less in distance, indicating the presence of obligatory unfolding intermediates (Fig. 1h,i)." Aren't the two statements contradictory to each other? How can a stochastic unfolding process show consistent unfolding peaks at similar distance? By looking at Fig. 1h, it is indeed stochastic. But, is Fig. 1i indeed representative?
7. The observation of TMEM16F trimer is striking. It has not been observed in any known TMEM16 structures. How to prove this is not an artifact? If using the reconstituted concatemer of TMEM16F, will the trimer still be observed? Are the trimers less dynamic or more stable than the dimers?
8. Since the authors have already tested TMEM16B in unfolding, it will be very informative if this pure ion channel without lipid scrambling activity, can be studied using HSAFM, especially for the dimer interface, dynamics of the monomers, and potential trimer formation.
9. As the authors introduced, TMEM16 monomers work independently from each other. It is unclear how dynamic movements/ structural heterogeneity of TMEM16F monomers are necessarily the cause of variations in rectification indexes. Please clarify. TMEM16 channels are notoriously subject to channel rundown especially under high calcium. Not sure the authors chose which time points to do the quantification. Can this complicate the patch clamp measurement? Please clarify.

Minor comments:

-1mM Ca, room temp, liposome reconstituted systems are about the same as the CryoEM sample preparation. Therefore, the statement that this is a key advantage of the approach is an over claim.
-Saying the data is "uncovering the full range of conformational changes of TMEM16F:" is overclaiming.

Reviewer #2:

Remarks to the Author:

The TMEM16x protein family is formed by members that function as ion channels while others serve as lipid scramblases and non-selective ion channels. In spite the advances that have been made in our understanding of the biophysics of ion and lipid transport in TMEM16x proteins, the conformational changes that occur in conjunction with these transport properties remain incompletely defined. This study focuses on the TMEM16F protein and provides strong evidence to show an unexpectedly large range of conformations that this protein can acquire and that these are related to the ability of the protein to mediate lipid and ion transport. The study is conducted with extreme care and the interpretation of the results is generally well justified. The constructive comments highlight areas that may require some clarification or some additional experimentation to increase the strength of the arguments presented even further.

- Introduction, page 3, second paragraph: Here the authors provide a critical analysis of the limitations of cryo-EM studies and the possible bias towards the more thermodynamically favourable conformations which is associated with the approach. In the following paragraph, the authors noted that techniques such as AFM and SMFS can allow monitoring of the broader populations of conformations that the protein may acquire. While I found this overall part of the Introduction well-written and objective, I would advise the authors to elaborate more fully on the limitations associated with the single molecule methods they have employed (see also comment below).

- Results, page 5: The rationale for using the NG108-15 cells for the determination of mechanical unfolding of TMEM16F should be explained in slightly greater detail. As shown in Fig. S2a, this cell line expresses both TMEM16F and B. A cell model of this kind may therefore lead to some false-positive events (related to the fact that it may be difficult to selectively assess TMEM16F unfolding - any membrane protein, especially TMEM16B which is homologous to TMEM16F and highly expressed in these cells, could be picked up by the probe). I appreciate that the authors also utilised transfected tagged TMEM16F to assess TMEM16F unfolding more specifically. However, the possibility that TMEM16F may form heteromers with TMEM16B, contributing to the heterogeneity of the signal observed, was not considered. This could be tested by using cells that do not have an endogenous expression of the TMEM16F or B and that could therefore provide a better system for more selective heterologous expression of either TMEM16F or B one at a time.

I also have some reservations on the immunostaining presented in Fig. S2. The signal for TMEM16B is impressively high and it does not seem to localise on the plasma membrane, but to have a broad distribution within the cell. Indeed, a similar pattern is observed for TMEM16F. I appreciate that subcellular localisation may be hard to dissect with epifluorescence microscopy, but the broad signal distribution observed in Fig. S2a could in part be the result of some non-specific staining. To probe for this, some control experiments should be performed in the absence of the primary antibody (to assess the specificity of the secondary antibody), and ideally in cells in which the target is deleted (either genetic knockout or siRNA mediated reduction in gene expression) to test for specificity of the primary antibody.

- Pg.8. start of section: The authors explained that "dozens of F-D spectra from N2B tagged TMEM16F obtained in identical conditions were aligned, superimposed, and then displayed as density plots". Could the authors clarify what determines an appropriate number of spectra to be considered in these studies? Were power calculations performed to define this number?

- Concerning the force vs- distance curves (e.g. Fig 1), I wonder if it may be useful to estimate the energy dissipated in each jump and correlate this with the energy needed to break the interactions between protein chains (e.g. hydrogen bridges). Doing this may strengthen the argument that the technique offers a direct assessment of folding.

- Pg 11, middle of the page: The authors noted that "due to the strong mechanical coupling between the monomers, TMEM16F might be occasionally unfolded in tandem, as the concatenation of two monomers.". I wondered whether this idea could be tested more fully by using TMEM16F concatemers (i.e. two TMEM16F subunits that are joined together by a linker)?

- Pg. 15, Conformational changes in TMEM16F upon Ca²⁺ binding: In these experiments, a high Ca²⁺ concentration was used (2 mM), which is well above the concentration (~100 μ M) which will already lead to max TMEM16F activation. I wonder if the high Ca²⁺ concentration used in these experiments might potentially lead to some surface charge screening effects which might potentially affect the unfolding pattern? To test for this, it may be useful to study the effect of Ca²⁺ on TMEM16F mutants in the main binding sites with reduced Ca²⁺ sensitivity. Any effect of Ca²⁺ on the unfolding of these mutants would be indicative of indirect effect of Ca²⁺ on unfolding. I invite the author to comment on these possibilities.

- Pg 19, electrophysiology studies: Here the authors noted that the variance of the measurements of the rectification index or of the ΔE_{rev} are larger for TMEM16F than for B and interpreted this as an indication that TMEM16F possesses a greater conformational heterogeneity. I have some suggestions for the authors to test this idea more fully:

(i) It may be expected that at a lower (i.e. non saturating) intracellular Ca²⁺ concentration, the variation in these parameters for TMEM16F may increase even further because the system would "move away" from the conformations promoted by saturating Ca²⁺. Thus, the ephys experiments of Fig. 5 should be repeated at a Ca²⁺ concentration near to EC₅₀ to test this possibility.

Another related experiment could involve measurements of variance of the rectification index and ΔE_{rev} in the presence of PIP₂, which again could bias the conformations acquired by TMEM16F towards PIP₂-bound conformations, thus potentially reducing variability of the electrophysiological parameters under consideration.

(ii) The larger patch-to-patch variability in ΔE_{rev} observed for TMEM16F compared to 16B is especially interesting as it may presumably indicate that different gating modes are promoted in different patches and that the various conformations in TMEM16F may differ in their ion selectivity and possibly even conductance. To test this idea more fully, the autocovariance or power spectra of tracts of stationary TMEM16F currents could be assessed in different patches to observe whether the cut-off frequencies of the spectra are consistent with populations of channels with different gating modes, and how this compares with TMEM16B. In addition, nonstationary noise analysis and standard variance-mean current plots could be constructed to verify whether there is larger variability in the calculated P_o and i in patches expressing TMEM16F (compared to TMEM16B). Although I realise that these detailed studies may be better reserved to a follow up paper, the authors may wish to comment on these possibilities.

(iii) The currents in figures 5a-d are presented normalised for the current measured at +100 mV. This mode of display is certainly useful because it enables visual comparison of the various traces, when current amplitude may vary from patch to patch. In the text, however, it will be important to state the actual mean current amplitude in pA, nA (and its variance) for TMEM16F and B and for patches excised from non-transfected cells. HEK293T cells may possess endogenous Ca²⁺ activated channels which could mediate a sizable current especially when Ca²⁺ is high (2 mM). Presenting these data is important because the magnitude of the TMEM16x current relative to the endogenous current may significantly influence the quantification of E_{rev} and rectification index; the interference of the

endogenous current would be more prominent for the TMEM16x current of lower amplitude.

Reviewer #3:

Remarks to the Author:

The article entitled "Structural heterogeneity of the TMEM16F channel" by Ye and co-authors describes the conformational dynamics of the TMEM16F scramblase protein under physiological conditions, by using a combined approach of fluorescence microscopy, electrophysiology, and, mainly, atomic force microscopy force spectroscopy (AFM-FS) and high-speed imaging (HS-AFM).

The manuscript is well written and delivers a convincing message. It is built upon a significant number of experiments and corresponding data analyses and interpretations. The results from AFM-FS and HS-AFM indicate that TMEM16F exhibits conformational heterogeneity at the dimerization interface, expanding cryo-EM knowledge. Compact TMEM16F dimers with high unfolding work vs. loose TMEM16F dimers with low unfolding work are described. HS-AFM imaging gives convincing topography as compared with AFM-simulated topography from cryo-EM structure. It also adds great value for the visualization of the dynamic change in protein and lipid topography when calcium is added.

The major concern in the manuscript is the lack of methodological details and the lack of a control experiment for AFM-FS experiments. From Fig. 1c (80 nm height range) one can see that such membranes are more complex (or "dirty") than a typical reconstituted supported lipid bilayer. Probing the inner membrane of a living cell with a non-specific tip with such a high threshold force will bring a lot of noise from other proteins, even if TMEM16F is over-expressed. Filtering of the raw AFM-FS data to get only the relevant information on the protein of interest only seems, at least, a very tricky thing to do.

- From ref. 54, 95 % of curves shown no binding, 3 % membrane tethers, and 2 % sawtooth-like patterns. These sawtooth-like patterns displayed different lengths and patterns because of the wide varieties of pulled proteins.

First the authors should describe better than "Processed traces were filtered and clustered according to ref.56, and their identification validated through the N2b tag as described (54)" in order for the reader to be able to repeat the experiment herein presented. Explain the steps with the specificities of the algorithm for TMEM16F. Also, give the output: how many (absolute number, %, etc...) of curves were considered for each experiment.

- One could also be concerned that the filtering might be affected by the variable topography and elasticity of the native membrane and as such by the uncertainty in the zero distance of the FD curve. Can you comment on this?

- Finally, and more importantly, there need to be a reasonable control experiment. This could for instance be a knock-out of the protein in native cell membranes. This could also be AFM-FS performed on a reconstituted system (supported lipid bilayer + protein). Or any other meaningful control experiment.

There are also few other minor points to address:

- Please explain in the text the added value of pulling from the inner membrane of an unroofed cell vs. pulling from the outer membrane of a living cell.

- From ref. 56: "We should also underline that the method is not designed to distinguish different unfolding pathways of the same protein. The filtering and the clustering procedure are by far too coarse for this scope". What about the fact that TMEM16F can be pulled by different subunits first, did you discard these curves?

- Fig 2c: did you test normality of the data to show a gaussian fit? The blue histogram in particular does not look normal. Maybe a box/violin plot is more adapted.

- Fig. 3e & video 1: there is no reason for this conformational change unless some energy is added to the system. Does this energy come from the AFM (tip scanning, laser temperature)?
- Please change "dozens" for the actual number of spectra.
- SMFS method: please mention how the InvOLS is calibrated, not just how the spring constant is.
- Fig. 4e & video 2: could this behaviour be a partial unfolding?
- Fig. S2c: the glass coverslip is depicted like a tipless AFM cantilever, that is very misleading as a first read. Can you change it?

REVIEWER COMMENTS

Reviewer #1 (Remarks to the Author):

Overall assessment:

Ye, et al offer a unique and intriguing addition to the TMEM16F structure/function literature, providing alternative evidence to attempt to bridge seemingly incompatible findings concerning TMEM16F substrate permeation and gating. Using primarily AFM, the authors find TMEM16F occupies a broader conformational landscape than those previously identified in cryo-EM structures. In particular, they find variability in the dimerization interface between monomers, where they show “slid” and “open/rotated” configurations using high speed AFM imaging. Their data support previous findings of subunit asymmetry in the dimer and that monomers function independently from one another. While the experimental approach is unique and informative, the following issues need to be further addressed or clarified.

Reply: We thank the reviewer for this overall positive assessment of our work

Major comments:

1. Title: TMEM16F is a scramblase and channel. The authors discussed both channel and scramblase activities. Just emphasizing channel in the title seems to be biased.

Reply: We thank the reviewer for pointing this out. In essence, we believe that a scramblase can be viewed as a lipid channel. In the revised version we have changed the title to better stress both lipid, and ion channelling functions of TMEM16F. The revised title now reads as “Structural heterogeneity of the ion and lipid channel TMEM16F”.

2. Cell line chosen: The authors used NG108-15 cell line, which highly expresses endogenous TMEM16B and TMEM16F (Fig. S2a). It was not justified why chose this cell line to overexpress the N2B constructs instead of using a cell line deficient of TMEM16F and B?

Reply: We thank the reviewer for his/her comments. In the revised version of the MS (first paragraph of the Result section) we have tried to better motivate our rationale for using the NG108-15 cell line, which can be summarized as follows:

- In a previous study from our group, we demonstrated that FS-AFM can be used to identify the most abundant protein expressed in isolated cells or cultured cell lines. Among the different proteins identified and cells considered, TMEM16F from NG108-15 cells could be recognized with the highest confidence and therefore this pair was chosen for follow-up investigations.
- It is hard to find a cell line that is deficient in all members of the anoctamin family. For instance, TMEM16F is expressed in most of the cell lines used, including HEK293, HeLa, and U87 (<https://doi.org/10.1074/jbc.m117.803049>,

<https://doi.org/10.1152/ajpcell.00228.2012>,
<https://doi.org/10.2147/OTT.S211725>, <https://doi.org/10.1242/jcs.217034>), so to validate our initial findings, we chose to overexpress a tagged TMEM16F directly in NG108-15 cells. We reasoned that overexpression of TMEM16F, which carries the N2B fingerprint, would allow us to dilute and filter out possible spurious events. Furthermore, the success rate of cell unroofing and membrane isolation procedure in NG108-15 cells is remarkably high (~80%), making this cell line a suitable platform for FS-AFM-based investigations, further justifying this choice (please refer to the next point).

- We also attempted to work with HEK293, which express TMEM16F but not TMEM16B and may therefore be more suitable for studying TMEM16F (<https://doi.org/10.1152/ajpcell.00228.2012>). Unfortunately, these cells did not adhere well to culture coverslips, preventing us from successfully isolating membrane fragments for AFM studies (<https://doi.org/10.1016/j.bbamem.2018.09.019>).

Will the heterologously expressed proteins coassemble with the endogenous TMEM16 proteins to form hetero-dimer (one endogenous subunit, one N2B subunit)?

Reply: We cannot definitively rule out the possibility that recombinant N2B-TMEM16F might co-assemble with native TMEM16s expressed by NG108-15 cells. However, it is important to notice that the analysis of TMEM16F unfolding spectra collected from non-transfected cells and cells overexpressing the recombinant N2B-TMEM16F construct yielded similar results, indicating mechanical/structural heterogeneity in both cases. Specifically, the unfolding patterns observed in TMEM16F from both non-transfected cells and cells overexpressing the N2B-TMEM16F construct are very similar and feature consistent force peak positions (see Fig. R1a,b (later on), and Supplementary Fig. S8). Likewise, the distribution of unfolding work for endogenous TMEM16F and the recombinant N2B construct show comparable bimodal distributions (Fig. [R1c]). Furthermore, the existence of heterotypic interactions between the C- and N-terminal domains was originally deduced from the analysis of tandem unfolding events observed in non-transfected cells (Fig. 3a-d and Fig. S7a-c). Thus, although the formation of heterodimers (one endogenous subunit, one N2B subunit) in transfected cells cannot be excluded, control experiments using non-transfected NG108-15 cells lacking the N2B-TMEM16F construct, as well as proteoliposomes embedding only purified TMEM16F (see below) clearly indicate that the conclusions drawn are valid and not affected by this possibility. We have revised the text (pages 9 and 10) to reflect some of these observations and have included the new data and analyses presented in Fig. R1 in the new Supplementary Figure S5 and Fig. 2 of the revised manuscript.

Is it possible that the expressed TMEM16F form heterodimer with endogenous TMEM16B? What is the evidence that the heterodimerization does not exist?

Reply: We appreciate the reviewer's inquiry into the potential heterodimerization of TMEM16F with endogenous TMEM16B. We would like to highlight that the available literature strongly supports the idea that TMEM16F and TMEM16B do not typically form heterodimers (<https://doi.org/10.1073/pnas.1303672110>). While it is true that the absence of evidence in published research is not definitive proof of absence, the prevailing evidence suggests that such interactions are presumably rare.

If it indeed exists, will it complicate the interpretation of the AFM results?

Reply: We agree that the presence of heterodimers or TMEM16B dimers could potentially impact the interpretation of the AFM data. For this reason, we performed additional FS control measurements using purified mTMEM16F reconstituted into artificial membrane bilayers, so where only mTMEM16F was present. Although gleaning an adequate number of force-distance curve from this preparation turned out to be very challenging due to the low reconstitution yields, the data obtained recapitulated many of the features observed in TMEM16F unfolded from cell membrane fragments. For instance, force peaks occurred at similar contour length among the three studied constructs: endogenous, N-N2B- and reconstituted mTMEM16F (Fig. R1a,b). Additionally, a comparable variability in unfolding work (as evidenced by similar standard deviations values of 7.5, 8.4, and 8.7 aJ for N-N2B-, native-, and reconstituted TMEM16F, respectively) pointing to a bimodal distribution was observed in the datasets (Fig. R1c). Finally, we would like to note that if a substantial proportion of unfolding spectra from non-transfected cells were due to spurious events like the unfolding of TMEM16F heterodimers or TMEM16B dimers, we would expect a significant reduction in heterogeneity when such events are diluted out by TMEM16F overexpression and N2B-based data filtering. Such reduction in heterogeneity was not observed. We report these new data in Fig. 2 and Supplementary figure S5 and detail the above thoughts in the Result section (pages 9,10).

Overall, these control experiments further support the validity of our conclusions and suggest that the interpretation of our data was unlikely to be affected in any significant way by spurious unfolding events.

Fig R1. (a) Sketches of the three constructs used in the SMFS experiments. (b) Density plots of $n=78$, $n=101$ and $n=68$ FD curves (left to right) of the constructs shown in a. (c) Violin plots of the calculated work to unfold the constructs in a. Average work is 16.4, 18.4, 20.4 aJ (left to right) with standard deviation of 7.5, 8.4 and 8.7 aJ, respectively. (The arrows highlight the presence of two dominant populations in all constructs, according to the unfolding work).

3. Sample preparation: In the unfolding experiments, how tightly the extracellular loops of TMEM16F is anchored to the coverslips? The authors mentioned that the maximal contour length of ~ 370 - 390 nm (Fig. S3) is roughly corresponding to the unfolding of the whole TMEM16F (Fig. 1e). However, if the membrane and protein partially detach from the coverglass, the F-D is no longer reliable. Please clarify/explain.

Reply: To what extent the support influences the results obtained by SMFS of adsorbed membranes containing proteins has been a long-standing concern in the field. A fundamental step towards the clarification of this concern was made by DJ Muller and coworkers in 2015 (<https://doi.org/10.1021/acs.nanolett.5b01223>) when they performed SMFS on membrane proteins (with 6 loops) from free standing membranes spanning nanoscopic pores. In this work they found that the unfolding and stability of bacteriorhodopsin shows no significant difference between freely

spanning and directly supported purple membranes assessing that the surface adsorption effects are negligible compared to instrumental noise even for a beta-sheet folded loop (which usually unfolds at higher forces).

The reviewer is also right about the possibility that — in principle — the membrane could detach from the surface while pulling a protein and therefore the detachment could produce unreliable curves. However, in practice this is something that empirically is not frequent at all. In the same paper mentioned above (<https://doi.org/10.1021/acs.nanolett.5b01223>, see Fig. 3) they modelled the effect of the elasticity of the membrane (i.e. if the membrane is not supported it bends due to the pulling force of the AFM tip) combined with the unfolding of a protein. They modelled the membrane bending with a cubic correction factor that properly fits the data. If the cubic correction factor for membrane elasticity is not considered, the apparent persistence length of the Worm Like Chain fit will be smaller than 0.1 nm, which would be very evident if such detachment happened (standard persistence length is 0.4 nm). We don't observe this persistence length deviation in our data.

4. Please also introduce what are the rational and evidence that a membrane protein embedded in the membrane has to unfold stepwise according to the order of the sequence. This is critical for assigning the F-D peaks to different TMs and interpreting the conformational changes of TM3, 4, 6 in Figure 4.

Reply: There are two major classes of protein constructs studied with AFM: tandems of globular proteins and membrane proteins.

In tandems of globular proteins, an extreme of the chain is attached to the surface and the other extreme is attached to the AFM tip. The sequence of unfolding is not defined by the position of the globule in the chain because the tension produced by the AFM tip retraction is constant at the extremes of each globule. Therefore, the sequence of unfolding is stochastic (please refer to the Fig. R2 and the reference shown below).

Fig. R2: The movement of the piezoelectric positioner stretches the protein until the applied force triggers the unfolding of a globule, increasing the contour length of the protein and relaxing the cantilever back to its resting position. The process is repeated until all domains are fully unfolded or the protein detaches from the tip/surface. The characteristic force–

extension sawtooth pattern curve from stretching a polyprotein is shown at the bottom (Image taken from <https://doi.org/10.1074/jbc.R700050200>)

In membrane proteins the mechanics is different. The tension produced by the AFM tip retraction is applied to the first folded domain that is still anchored on the membrane, and not necessarily to the rest of the folded subdomains that are still embedded in the membrane. It is certainly possible to unfold multiple subdomains at once, but it is not possible to unfold a sub-domain down in the sequence without having unfolded the previous ones, because the tension could not have reach it (Fig.R3).

Fig. R3: Cartoon depicting the sequential unfolding and extraction of an individual bacteriorhodopsin molecule (Image taken from <https://doi.org/10.1126/science.aam8370>)

Evidence for this sequential unfolding is reported in many articles about membrane proteins (e.g. <https://doi.org/10.1126/science.aah7124>; <https://doi.org/10.1038/nchembio.2169>; <https://doi.org/10.1126/science.288.5463.143>, and many others) so peak assignment is straightforward if the length of the F-D curves corresponds to the expected one.

In terms of peak assignment, the reviewer might also be concerned about a configuration where the tip binds a loop instead of the C- or N-terminus of the protein (please refer to the Fig. R4).

Fig. R4: Cartoon depicting the unfolding of a membrane protein grabbed from an intervening loop connecting two contiguous transmembrane helices (Image taken from <https://doi.org/10.7554/eLife.77427>).

However, in this case the unfolding curve will be much shorter, so such a F-D curve will not be considered in our analysis.

5. Why there are only a few unfolding events recorded, given the large folded the structure at the cytosolic domain? Does this imply the cytosolic N-terminus cannot unfold?

Reply: Fig. 2a and Supplementary Table 1 indicate that unfolding events at around 0 nm in tip-sample distance are not very frequent and, when they happen, the unfolding force is usually lower than 50 pN. These findings suggest that the forces necessary to stretch the cytosolic N-terminus (NCD) are significantly lower compared to the tension necessary to extract and unfold the transmembrane helices from the bilayer. Overall, this observation is in contrast with a highly mechanically stable cytosolic domain. Indeed, cryo-EM structures feature several poorly defined loops connecting short secondary elements within the large cytosolic N-terminal domain (Supplementary Fig. S12) which likely confer mechanical instability (in the revised version of Fig. S12 we have annotated these unstructured regions on the primary sequence of TMEM16F). Consistently, we found that its unfolding occurs at lower tensions and that the position of force peaks varied considerably (see Fig. 4a-d). Further evidence supporting the notion that the bulky N-terminus might be rather flexible comes from HS-AFM imaging, which indicates significant differences in protrusion heights of the NCD from the membrane plane, of ~3 and ~4nm (Fig. 4f).

TMEM16s have 10 TMs, why usually only 5-6 peaks were observed in the F-D curves?

Reply: To the best of our knowledge, there is no general framework that allows to predict the number of major force peaks during the unfolding of a membrane protein. A major force peak occurs in the presence of a well-defined energy barrier in the energy landscape of the protein. To date, there is only empirical evidence of the occurrence of force peaks along the sequence from 11 different membrane proteins (see a collection of these data in the following swarm plot from <https://elifesciences.org/articles/77427/figures#fig4>).

Fig. R5. *Conditional probability for the occurrence of unfolding peaks extracted from SMFS literature (see <https://doi.org/10.7554/eLife.77427>). Unfolding peaks occurs most likely in the loops (82%) than in transmembrane domains ($n_{\text{peaks}} = 54$, from 11 SMFS experiments of different membrane proteins). The points in the green (yellow) region represents unfolding peaks occurred in a cytoplasmic (extracellular) loop, the point in the pink area occurred in a transmembrane domain. The points above the green and below the yellow regions occurred in cytoplasmic and extracellular domains, respectively. The scale is approximate because in rare occasion loops are longer than 10 nm.*

These data show that, during unfolding, 82% of the force peaks are obtained from loop regions connecting the transmembrane helices (Fig. R5), mostly from the cytoplasmic side. In these reports, transmembrane domains are also unfolded pairwise, which can give a rough estimate for the shape of the energy landscape of the protein. TMEM16s behave similarly to those other membrane proteins.

6. Page 8, The authors wrote “the unfolding process and the occurrence of force peaks are stochastic in nature”. However, in Page 6, the authors described “a reproducible sequence of force peaks was observed for both, WT-16F and N-N2B-16F with peaks varying much in amplitude but less in distance, indicating the presence of obligatory unfolding intermediates (Fig. 1h,i).” Aren’t the two statements contradictory to each other? How can a stochastic unfolding process show consistent unfolding peaks at similar distance? By looking at Fig. 1h, it is indeed stochastic. But, is Fig. 1i indeed representative?

Reply: We thank the reviewer for this comment and agree that the two sentences might have been a source of potential confusion to the reader. The term “reproducible” in the text referred to a certain degree of regularity or consistency in the unfolding process that can be observed across different molecules and did not mean that the unfolding process is completely deterministic and predictable. Thus, the term “reproducible” refers to the underlying probability distribution of the force peaks (see Fig. 4a,c), while the term “stochastic” refers to the specific sequence of force peaks observed during a single unfolding event and to the unfolding force at which they occur. Fig. 1h highlight the reproducibility of the unfolding process by showing that, despite this variability, there are certain obligatory intermediates that are consistently formed (such force peaks have now been marked, see also Yu et al., 2017 <https://doi.org/10.1126/science.aah7124> where obligatory intermediates are highlighted in the unfolding of membrane proteins) leading to some similarities in the pattern of force peaks across different unfolding events. In the revised MS we have modified the text and Fig. 1h to better stress this point and reduce possible ambiguities.

7. The observation of TMEM16F trimer is striking. It has not been observed in any known TMEM16 structures. How to prove this is not an artifact? If using the reconstituted concatemer of TMEM16F, will the trimer still be observed? Are the trimers less dynamic or more stable than the dimers?

Reply: We thank the reviewer for this appropriate comment. The reviewer is certainly correct and currently, all the available structures of TMEM16 family members are

dimeric. Yet, circumstantial findings suggesting the presence of higher order assemblies were occasionally reported in the literature. For instance, in <https://doi.org/10.1038/nature09583>, western blot analysis presented in Fig. 2b suggests the presence of multimers alongside TMEM16F monomers in Ba/F3 cells overexpressing TMEM16F. Likewise, findings hinting to higher order oligomers were also reported for TMEM16A channel (Fig. 7a in <https://doi.org/10.1074/jbc.M110.174847>). In regard to the reviewer's suggestion to attempt reconstitution and imaging of concatemers, we did not try to pursue this quite laborious direction. Because subunits are covalently linked in this construct and each contains an independent lipid and ion permeation pathway, we would expect to find trimers in such experiments. Nonetheless, observation of trimers would not constitute evidence of a native trimeric assembly as one can still argue that such structure was imposed by the introduced linkage of the three polypeptide chains. In our study observation ($n=2$) and unfolding ($n=3$) of TMEM16F trimers were actually very rare, and therefore the interpretation of these data remains uncertain. However, also in light of a recent study from Scheurings' group, demonstrating the switching of tetrameric TRPV3 channels to a non-canonical pentameric arrangement, we believe these data showing unorthodox oligomerization states of TMEM16F are still worth discussing to some extent (<https://doi.org/10.1038/s41586-023-06470-1>). We revised the text to mention these recent findings (page 13).

8. Since the authors have already tested TMEM16B in unfolding, it will be very informative if this pure ion channel without lipid scrambling activity, can be studied using HSAFM, especially for the dimer interface, dynamics of the monomers, and potential trimer formation.

Reply: The reviewer is correct, and imaging of TMEM16B would certainly be of value and provide important complementary information. However, HSAFM imaging of native membrane specimens with sub-molecular resolution is very challenging, and even if conditions could be found to reach the instrument's resolution limit (in X,Y dimensions) of ~ 1 nm, it would still be very hard to unambiguously identify individual TMEM16B among the many membrane proteins expressed. Consequently, thus far, the vast majority of the HSAFM imaging studies of membrane proteins have been performed on purified and reconstituted samples rather than isolated membrane fragments (<https://doi.org/10.1016%2Fj.sbi.2019.02.008>). Unfortunately, we have not yet established overexpression, purification, and reconstitution of TMEM16B channels into liposomes, and although we are working in this direction, there is still a considerable way to go.

9. As the authors introduced, TMEM16 monomers work independently from each other. It is unclear how dynamic movements/ structural heterogeneity of TMEM16F monomers are necessarily the cause of variations in rectification indexes. Please clarify. TMEM16 channels are notoriously subject to channel rundown especially under high calcium. Not sure the authors chose which time points to do the quantification. Can this complicate the patch clamp measurement? Please clarify.

Reply: We never claimed that there is a relation of "causal necessity" between the observed dynamic movements/ structural heterogeneity and the variation of

rectification indexes. We provide evidence that some functional properties, such as current-voltage relationship, have a higher variability in TMEM16F than in TMEM16B, and that such variability is related to structural/mechanical heterogeneity (Fig. 5). Specifically, we show that according to the unfolding work there are two prominent subpopulations of TMEM16F and overall larger data dispersion compared to TMEM16B. Thus, a correlation between variability in unfolding work and some electrophysiological quantities exists. We agree with the reviewer that these structural fluctuations are not necessarily the cause of the observed functional variability, but according to the structure-function paradigm this is a reasonable working hypothesis. Thus, if we were to assume that the two TMEM16F populations differ in their degree of voltage-dependency/rectification, depending on the specific mix of channels present in the inside-out patch, the degree of current rectification measured would vary across the different experiments. This variability is expected to reduce in TMEM16B, which display a narrower distribution of unfolding work. However, providing a precise mechanism connecting the structural heterogeneity of TMEM16F to its electrophysiological properties (i.e. how the observed variability in protein mechanical properties or dimerization interface affect the conformation of the permeation pathway and ion movements in response to an applied electrical field) is beyond the scope of this paper and might be better addressed by other techniques. We have slightly revised the last paragraph of the result section to clarify.

To reduce the possible error due to the current rundown, we performed the experiments after the fast component of run-down when the current reach reasonable steady-state, taking into account the length of the recordings. We observed similar variability even when TMEM16F was activated with lower Ca^{2+} concentrations, suggesting that the results are not substantially affected by rundown.

Minor comments:

-1 mM Ca, room temp, liposome reconstituted systems are about the same as the CryoEM sample preparation. Therefore, the statement that this is a key advantage of the approach is an over claim.

Reply: We only partially agree with the reviewer on this specific point. Cryo-EM studies are performed on frozen samples at cryogenic temperatures, whereas AFM experiments are performed at RT, in a liquid environment, and at ambient pressure. Consequently, only the latter technique allows direct observation of protein conformational dynamics and function simultaneously. In addition, cryo-EM relies heavily on ensemble averaging and data processing for structural determination. This means that low-probability states can easily be missed and discarded. AFM, on the other hand, is a truly single-molecule method and, in principle, allows the observation and manipulation of individual proteins and even the characterization of rare states. We believe that these are the key advantages over cryo-EM that allow us to capture the structural dynamics and heterogeneity of TMEM16F. Of course, like all techniques, AFM comes with its own limitations, which we briefly outlined in the discussion of the revised MS.

-Saying the data is “uncovering the full range of conformational changes of TMEM16F.” is overclaiming.

Reply: we have revised the text as follow: “Our SMFS and HS-AFM results extend previous in vitro structural findings to native membranes, and unveil a range of transitions in the quaternary assembly of TMEM16F, revealing previously unknown large-scale conformational dynamics.”

Reviewer #2 (Remarks to the Author):

The TMEM16x protein family is formed by members that function as ion channels while others serve as lipid scramblases and non-selective ion channels. In spite the advances that have been made in our understanding of the biophysics of ion and lipid transport in TMEM16x proteins, the conformational changes that occur in conjunction with these transport properties remain incompletely defined. This study focuses on the TMEM16F protein and provides strong evidence to show an unexpectedly large range of conformations that this protein can acquire and that these are related to the ability of the protein to mediate lipid and ion transport. The study is conducted with extreme care and the interpretation of the results is generally well justified. The constructive comments highlight areas that may require some clarification or some additional experimentation to increase the strength of the arguments presented even further.

Reply: We thank the reviewer for this overall positive assessment of our work and the constructive feedback provided.

- Introduction, page 3, second paragraph: Here the authors provide a critical analysis of the limitations of cryo-EM studies and the possible bias towards the more thermodynamically favourable conformations which is associated with the approach. In the following paragraph, the authors noted that techniques such as AFM and SMFS can allow monitoring of the broader populations of conformations that the protein may acquire. While I found this overall part of the Introduction well-written and objective, I would advise the authors to elaborate more fully on the limitations associated with the single molecule methods they have employed (see also comment below).

Reply: We have revised the discussion (first paragraph) to acknowledge some of the limitations of the methods used, including the restriction of AFM imaging to surface structures and uncertainties related to protein-probe interactions within complex native cellular membranes.

- Results, page 5: The rationale for using the NG108-15 cells for the determination of mechanical unfolding of TMEM16F should be explained in slightly greater detail. As shown in Fig. S2a, this cell line expresses both TMEM16F and B. A cell model of this kind may therefore lead to some false-positive events (related to the fact that it may be difficult to selectively assess TMEM16F unfolding - any membrane protein, especially TMEM16B which is homologous to TMEM16F and highly expressed in these cells, could be picked up by the probe). I appreciate that the authors also utilised transfected tagged TMEM16F to assess TMEM16F unfolding more specifically. However, the possibility that TMEM16F may form heteromers with TMEM16B, contributing to the heterogeneity of the signal observed, was not considered. This could be tested by using cells that do not have an endogenous expression of the TMEM16F or B and that could therefore provide a better system for more selective heterologous expression of either TMEM16F or B one at a time.

Reply: We agree with the reviewer. In the first paragraph of the revised Result section, we try to better motivate the rationale behind using NG108-15 cell lines. Additionally, we refer the Reviewer to our reply to Reviewer 1 (second comment), who also brought up very similar concerns. To rule out that the presence of heterodimers or other spurious events might contribute to the observed heterogeneity we have performed additional unfolding experiments from purified TMEM16F reconstituted in artificial membranes, where TMEM16F was the sole protein present. The analysis of these new data indicates that the three studied constructs (endogenous, N-N2B- and reconstituted mTMEM16F (Fig. R1a,b)) have a comparable variability of unfolding work (as evidenced by similar standard deviations values of 7.5, 8.4, and 8.7 aJ for N-N2B-, native-, and reconstituted TMEM16F, respectively) and similar distributions with two dominant populations (Fig. R1c). Finally, we would like to note that if a substantial proportion of unfolding spectra from non-transfected cells were due to spurious events like the unfolding of TMEM16F heterodimers or TMEM16B dimers, we would expect a significant reduction in heterogeneity when such events are diluted out by TMEM16F overexpression and N2B-based data filtering. Such reduction in heterogeneity was not observed. We report these new data in Fig. 2 and Supplementary Figure S5 and detail the above thoughts in the Result section (pages 9,10)

Overall, these control experiments further strengthen the validity of our conclusions and suggest that the interpretation of our data was unlikely to be affected in any meaningful way by false positive events.

Fig R1. (a) Sketches of the three constructs used in the SMFS experiments. (b) Density plots of $n=78$, $n=101$ and $n=68$ FD curves (left to right) of the constructs shown in a. (c) Violin plots of the calculated work to unfold the constructs in a. Average work is 16.4, 18.4, 20.4 aJ (left to right) with standard deviation of 7.5, 8.4 and 8.7 aJ, respectively. (The arrows highlight the presence of two dominant populations in all constructs, according to the unfolding work).

I also have some reservations on the immunostaining presented in Fig. S2. The signal for TMEM16B is impressively high and it does not seem to localise on the plasma membrane, but to have a broad distribution within the cell. Indeed, a similar pattern is observed for TMEM16F. I appreciate that subcellular localisation may be hard to dissect with epifluorescence microscopy, but the broad signal distribution observed in Fig. S2a could in part be the result of some non-specific staining. To probe for this, some control experiments should be performed in the absence of the primary antibody (to assess the specificity of the secondary antibody), and ideally in cells in which the target is deleted (either genetic knockout or siRNA mediated reduction in gene expression) to test for specificity of the primary antibody.

Reply: We thank the reviewer for his/her relevant comment. Indeed, using epifluorescence microscopy to assess the subcellular distribution of endogenous

TMEM16s is suboptimal, as all focal planes will contribute to the fluorescence signal in focus, leading to a detrimentally high background level, thus preventing precise analysis of the TMEM16s localization.

For that reason, we addressed the subcellular distribution of the TMEM16F/B using confocal and super-resolution microscopy. Here we took advantage of the optical sectioning capabilities of confocal microscopy to precisely address this question.

In Figure R6, we addressed the question on the specificity of labelling by performing a negative control by incubating the sole fluorescent secondary antibody on NG108-15 cells and compared it to cells exposed to both primary and secondary antibodies. Under those conditions, we observed fluorescence levels in the control which were >10 times inferior to those of the standard immunofluorescence. Controls were conclusive both using a STAR RED-conjugated or a STAR GREEN anti-rabbit secondary antibody (Abberior GmbH).

Fig. R6: Immunostaining of neuroblastoma NG108-15 cells treated with Abberior STARGREEN secondary antibody in the absence of a primary antibody (control, left panel), primary antibodies against TMEM16B (middle), and TMEM16F (right panel), demonstrating that both membrane proteins are abundantly expressed. In all the panels DAPI was used as nuclear marker.

Next, we addressed the distribution of TMEMs in the cell (Figure R7). A STEDYCON system from Abberior Instruments was used for this purpose. In confocal mode, we used between 1 and 3 excitation lines (405nm, 488nm and 640nm). APD detectors available on the system having an intrinsically low background (<0.5 photons per pixel with a 2.5us pixel dwell time), the observed signal is predominantly due to the sample itself. All confocal and STED images were obtained with a pinhole-size of 64µm (1.18 Airy units @650nm). In both TMEM16F and B staining with Abberior STAR RED was dense and spot-like in xy (Fig. R7a) or xz section (Fig. R7b) and appeared to be excluded from the cytosol and sometimes, but not always, present in the nucleus, with a predominant contribution coming from the plasma membrane of the cell. To confirm the membrane distribution, we performed confocal scans in xz on fixed cells labelled for TMEM16F or TMEM16B, for cholesterol (Abberior Membrane STARRED) and nucleus (DAPI) (Fig. R7c). This imaging confirmed the colocalization of membrane STARRED with TMEM16F or B indicating that both proteins were

localized at the plasma membrane. We note that cell nuclei were prominently labelled with Membrane STARRED: this is likely due to the presence of cholesterol in the nucleus, as previously reported.

Fig. R7: Confocal analysis of the subcellular distribution of TMEM16F and TMEM16B. (a) XY confocal scan of neuroblastoma NG108-15 cells labelled in immunofluorescence with Abberior STAR RED for TMEM16F and DAPI reveals dot-like distribution of TMEM16F at the plasma membrane. (b) XZ sections of TMEM16F expressing cells labelled with Abberior STARRED and DAPI in immunofluorescence reveal exclusion of TMEM16F from cytosol and low signal in the nucleus. (c) XZ sections of NG108-15 cells immunolabeled with primary antibodies against TMEM16F (left column) and TMEM16B (right column), membrane (Abberior Membrane STARRED) and nucleus (DAPI). TMEM16s labelling was developed with Abberior STAR GREEN secondary antibody. Immunofluorescence reveals

colocalization of both proteins with the plasma membrane and confirm their exclusion from the cytosol. Occasional nuclear localization of TMEM16F or 16B was observed.

Finally, we investigated structural aspects of the fluorescent labelling of TMEM16F (Fig. R8) using STED microscopy, in order to check whether the diffraction-limited spots observed in confocal correspond to single molecules or aggregates. In Fig. R8, STED images reveal spots at the cell membrane with a full width at half-maximum in the range of 40-50nm (inset, Fig. R8), which corresponds to the size of channels labelled with single or double primary+secondary antibody complexes.

Fig. R8: Observation of TMEM 16F in stimulated emission depletion (STED) nanoscopy. Top: confocal observation of TMEM16F immunolabeled with Abberior STAR RED at the plasma membrane of NG108-15 cells reveal a dense, yet spot-like distribution of the protein at the cell surface. Bottom: STED nanoscopy at resolution <40nm confirms that TMEM16F is not aggregated at the plasma membrane, nor generates a continuous signal within the cell. Inset: Gaussian fit on one typical spot of TMEM16F reveals a size within the

resolution limit of the used STED microscope, which could be consistent with single TMEM16F molecules in complex with primary and secondary antibodies (the largest dimension of an antibody being 15nm).

Based on this evidence, we conclude that NG-108-15 cell line endogenously expresses TMEM16F and B, that these proteins are predominantly localized at the plasma membrane in the likely form of dimers, and that the immunofluorescence protocol used is specific. We report these new immunofluorescence data in Supplementary Figures S2a, S3 and S4 of the revised MS.

- Pg.8. start of section: The authors explained that “dozens of F-D spectra from N2B tagged TMEM16F obtained in identical conditions were aligned, superimposed, and then displayed as density plots”. Could the authors clarify what determines an appropriate number of spectra to be considered in these studies? Were power calculations performed to define this number?

Reply: To the best of our knowledge, there is no standard with a consensus on the number of FD curves that represents an 'appropriate' number. Our rationale for what is the minimum number of FD curves to present is based on the precision of the determination of the major unfolding intermediates (histograms in Fig. 4a and 4c). These global histograms show the presence of a distribution of contour lengths that accumulate around the most likely values that are then interpreted as obligatory intermediates (see Yu et al., 2017 <https://doi.org/10.1126/science.aah7124>, Maity et al., 2015, <https://doi.org/10.1038/ncomms8093>). The criterion that we adopt is to obtain a standard error of the mean of the position of these peaks at least ten times smaller than their separation (which is usually in the order of 20–40 nm). The standard deviation of these peaks is usually about 10 nm, therefore the minimum sample size according to our criterion is ~25 FD curves.

- Concerning the force vs- distance curves (e.g. Fig 1), I wonder if it may be useful to estimate the energy dissipated in each jump and correlate this with the energy needed to break the interactions between protein chains (e.g. hydrogen bridges). Doing this may strengthen the argument that the technique offers a direct assessment of folding.

Reply: Our unfolding experiments are fast processes, therefore far from the ideal case of an infinitely slow unfolding. The work W we calculate in Fig. 2b is only a semiquantitative estimate of the free energy ΔF of the folded protein, recalling Jarzynski inequality $\Delta F \leq \overline{W}$, thus our interpretation is only qualitative. A more robust method for assessing the protein's energy landscape using AFM would involve unfolding the protein at various pulling velocities and utilizing the Bell-Evans model to derive kinetic parameters. However, this necessitates achieving high pulling efficiencies. While this can be attained through a specialized preparation involving a high protein density per surface area, our current system—both in the native membrane and proteoliposomes—has not allowed us to reach this level of efficiency.

- Pg 11, middle of the page: The authors noted that “due to the strong mechanical coupling between the monomers, TMEM16F might be occasionally unfolded in tandem, as the concatenation of two monomers.”. I wondered whether this idea could be tested more fully by using TMEM16F concatemers (i.e. two TMEM16F subunits that are joined together by a linker)?

Reply: We thank the reviewer for this thoughtful comment. Indeed, we have directly tested and demonstrated this hypothesis in the past for the CNGA1 channel, for which a tandem construct (concatenating two CNGA1 polypeptides via a short 10 a.a. linker between the C-terminal end of the first and the N-terminal end of the second subunit) has been available since early 2000 and has since been extensively characterized (<https://doi.org/10.1007/s00249-008-0312-1>). The pulling of the CNGA1-CNGA1 tandem demonstrated that CNGA1 channels can indeed unfold as a sequence of two linked subunits, similar to what we observed for TMEM16F (<https://doi.org/10.1038/ncomms8093>). It is important to note that in the case of CNGA1 we did not detect any variability in the oligomerization interfaces, probably because CNGs are tetrameric, with a single, centrally located pore. Such engineered constructs are usually laborious to design and generate de novo, require careful functional characterization and, when working, typically suffer from much lower expression levels. Furthermore, in the case of TMEM16F the dynamics of the two subunits reported here could be dramatically constrained by the engineered linker, so for the purposes of this revision we prefer not to go into this laborious direction, also considering the revision timeframe. As further evidence for tandem unfolding, we report below (Fig. R9) and in the Supplementary Fig. S7 of the revised MS the pulling of a TMEM16F dyad from proteoliposomes, presumably in the C-C configuration. Nonetheless, we reserve to better address this in future studies, possibly by designing different concatemers with head-to-tail, head-to-head and tail-to-tail couplings.

Fig. R9. Reconstituted mTMEM16F occasionally unfolds in tandem (blue trace). The F-D curve aligns well with tandem traces from endogenous TMEM16F in the C-C configuration (black trace), where the two protomers associate via C-terminal interactions.

- Pg. 15, Conformational changes in TMEM16F upon Ca²⁺ binding: In these experiments, a high Ca²⁺ concentration was used (2 mM), which is well above the

concentration (~100 μM) which will already lead to max TMEM16F activation. I wonder if the high Ca^{2+} concentration used in these experiments might potentially lead to some surface charge screening effects which might potentially affect the unfolding pattern? To test for this, it may be useful to study the effect of Ca^{2+} on TMEM16F mutants in the main binding sites with reduced Ca^{2+} sensitivity. Any effect of Ca^{2+} on the unfolding of these mutants would be indicative of indirect effect of Ca^{2+} on unfolding. I invite the author to comment on these possibilities.

Reply: We thank the reviewer for this comment and the clever experimental suggestion. Interpreting the unfolding behaviour of known binding mutants such as E667Q might not be straightforward. Binding and gating interdepend on each other as a consequence of the principle of reciprocity (the so called binding-gating problem, <https://doi.org/10.1038/sj.bjp.0702164>) and thus the low apparent Ca^{2+} affinity observed in some of those mutants not necessarily reflect complete disruption of calcium binding (<https://doi.org/10.1016%2Fj.cell.2012.07.036>). Additionally, one must recall that in TMEM16F several acidic residues dispersed across helices TM6-TM8, TM2 and TM10 contribute to the Ca^{2+} primary and the regulatory binding sites. Hence, completely abolishing calcium coordination at such high concentration (2mM) would likely necessitate generating a multiple mutant, posing considerable experimental challenges (*e.g.* lack of proper folding and/membrane targeting, low expression levels, etc.). Here, instead of exploiting a loss-of-function mutant to demonstrate consistent unfolding in the presence and absence of calcium, we have taken an alternative route. Specifically, we performed unfolding experiments of the gain-of-function mutant Y563K in calcium-free solution and observed a comparable unfolding pattern to N2B-TMEM16F in 2mM Ca^{2+} (Supplementary Figure S9). Overall, these data indirectly suggest that surface charge screening is unlikely to have a substantial impact on both the structure and unfolding of TMEM16F. In support of this view, on the one hand functional measurements performed by other groups at high calcium concentrations (1 and 5 mM) show comparable calcium-activated ion and lipid transport activity to those observed at ~100 μM (<https://doi.org/10.1085/jgp.202012704>). On the other hand, the effect of different electrolytes, including calcium ions, on the nanomechanical properties of supported lipid bilayers has also been thoroughly assessed in previous force spectroscopy investigations. According to these earlier studies, the presence of 2 mM CaCl_2 (in a buffer containing 150mM NaCl) is not expected to significantly alter the gross mechanical resistance and plastic properties of supported planar membranes (<https://doi.org/10.3109/09687688.2013.868940>). Finally, from a more theoretical perspective, although charge screening may impact the membrane zeta-potential and to some extent the magnitude of rupture forces, our analysis primarily focused on comparing the contour lengths (*i.e.* polypeptide lengths) of the unfolded segments between the Ca^{2+} -bound and Ca^{2+} -free conditions. This comparison is predominantly influenced by the protein's membrane topology and 3D architecture and is expected to be relatively insensitive to local surface electrostatics.

- Pg 19, electrophysiology studies: Here the authors noted that the variance of the measurements of the rectification index or of the ΔE_{rev} are larger for TMEM16F than

for B and interpreted this as an indication that TMEM16F possesses a greater conformational heterogeneity. I have some suggestions for the authors to test this idea more fully:

(i) It may be expected that at a lower (i.e. non saturating) intracellular Ca^{2+} concentration, the variation in these parameters for TMEM16F may increase even further because the system would "move away" from the conformations promoted by saturating Ca^{2+} . Thus, the ephys experiments of Fig. 5 should be repeated at a Ca^{2+} concentration near to EC_{50} to test this possibility.

Reply: We performed additional experiments measuring the rectification index and ΔE_{rev} for TMEM16F activated at lower Ca^{2+} concentration (13 μM). We found that the variability of these parameters is similar to that observed with saturating Ca^{2+} concentration suggesting that TMEM16F could have similar structural flexibility both at low and high Ca^{2+} . We have included these results in the new Supplementary Figure S10 and provide them below for the convenience of the Reviewers (Fig. R10).

Fig R10. The variability of electrical properties in TMEM16F channels. (a-b) The inside-out excised membrane patches expressing TMEM16F were recorded in symmetric NaCl (140 mM, a) and lower intracellular NaCl (14 mM, b) solution. Their IV relations were determined via voltage ramps from -80 mV to +100 mV. Currents were activated by 13 μM CaCl_2 . (c-d) Comparison of the changes of rectification for TMEM16F activated by the indicated Ca^{2+} concentrations. Rectification was calculated as the ratio between currents measured at +60 and -60 mV ($|I_{+60}/I_{-60}|$) and normalized to the average rectification value (μ , $n=16$ for 1 mM and $n=8$ for 13 μM). (f) Comparison of the shift of reversal potentials after

the replacement of 140 mM NaCl with 14 mM NaCl normalized to the average value for TMEM16F activated by the indicated Ca^{2+} concentrations (n=19 for 1 mM and n=8 for 13 μM).

Furthermore, upon studying TMEM16B, we unexpectedly discovered a significant alteration in permeability, suggesting the possibility that TMEM16B becomes more sodium-permeable at low Ca^{2+} concentrations. Further experiments are necessary to fully clarify this issue, and we prefer to include these results in a separate paper.

Another related experiment could involve measurements of variance of the rectification index and ΔE_{rev} in the presence of PIP₂, which again could bias the conformations acquired by TMEM16F towards PIP₂-bound conformations, thus potentially reducing variability of the electrophysiological parameters under consideration.

Reply: We agree that the use of PIP₂ could give some information about the variability of TMEM16F mediated current. However, we used PIP lipids in the past obtaining very contrasting results, therefore we think that these data are more suitable for a dedicated paper.

(ii) The larger patch-to-patch variability in ΔE_{rev} observed for TMEM16F compared to 16B is especially interesting as it may presumably indicate that different gating modes are promoted in different patches and that the various conformations in TMEM16F may differ in their ion selectivity and possibly even conductance. To test this idea more fully, the autocovariance or power spectra of tracts of stationary TMEM16F currents could be assessed in different patches to observe whether the cut-off frequencies of the spectra are consistent with populations of channels with different gating modes, and how this compares with TMEM16B. In addition, nonstationary noise analysis and standard variance-mean current plots could be constructed to verify whether there is larger variability in the calculated P_o and i in patches expressing TMEM16F (compared to TMEM16B). Although I realise that these detailed studies may be better reserved to a follow up paper, the authors may wish to comment on these possibilities.

Reply: We agree that the analysis proposed by the reviewer could provide interesting insights on the possible role of different gating modes in controlling ion selectivity. However, this type of analysis requires a very long recording with steady state currents, which can be challenging to obtain due to run down especially for TMEM16F. We believe that such a sophisticated and demanding analysis is more suitable for a dedicated paper.

(iii) The currents in figures 5a-d are presented normalised for the current measured at +100 mV. This mode of display is certainly useful because it enables visual comparison of the various traces, when current amplitude may vary from patch to patch. In the text, however, it will be important to state the actual mean current

amplitude in pA, nA (and its variance) for TMEM16F and B and for patches excised from non-transfected cells. HEK293T cells may possess endogenous Ca²⁺ activated channels which could mediate a sizable current especially when Ca²⁺ is high (2 mM). Presenting these data is important because the magnitude of the TMEM16x current relative to the endogenous current may significantly influence the quantification of Erev and rectification index; the interference of the endogenous current would be more prominent for the TMEM16x current of lower amplitude.

Reply: We added the requested information in the legend of Figure 5.

Reviewer #3 (Remarks to the Author):

The article entitled "Structural heterogeneity of the TMEM16F channel" by Ye and co-authors describes the conformational dynamics of the TMEM16F scramblase protein under physiological conditions, by using a combined approach of fluorescence microscopy, electrophysiology, and, mainly, atomic force microscopy force spectroscopy (AFM-FS) and high-speed imaging (HS-AFM).

The manuscript is well written and delivers a convincing message. It is built upon a significant number of experiments and corresponding data analyses and interpretations. The results from AFM-FS and HS-AFM indicate that TMEM16F exhibits conformational heterogeneity at the dimerization interface, expanding cryo-EM knowledge. Compact TMEM16F dimers with high unfolding work vs. loose TMEM16F dimers with low unfolding work are described. HS-AFM imaging gives convincing topography as compared with AFM-simulated topography from cryo-EM structure. It also adds great value for the visualization of the dynamic change in protein and lipid topography when calcium is added.

Reply: We thank the reviewer for this overall positive assessment of our work and appreciation of HS-AFM imaging and efforts of structural reconstruction.

The major concern in the manuscript is the lack of methodological details and the lack of a control experiment for AFM-FS experiments. From Fig. 1c (80 nm height range) one can see that such membranes are more complex (or "dirty") than a typical reconstituted supported lipid bilayer. Probing the inner membrane of a living cell with a non-specific tip with such a high threshold force will bring a lot of noise from other proteins, even if TMEM16F is over-expressed. Filtering of the raw AFM-FS data to get only the relevant information on the protein of interest only seems, at least, a very tricky thing to do.

Reply: The reviewer is correct, filtering the AFM-FS raw data is a difficult task. We have been working on it for several years and have produced detailed routines for that goal (*Galvanetto et al., 2018, <https://doi.org/10.1016/j.bpj.2018.02.004>, Ilieva et al., 2020, <https://doi.org/10.1093/bioinformatics/btaa626>, Galvanetto et al., 2022 <https://doi.org/10.7554/eLife.77427>). These methods were benchmarked and perform reliably both on purified proteins and in native preparations.*

- From ref. 54, 95 % of curves shown no binding, 3 % membrane tethers, and 2 % sawtooth-like patterns. These sawtooth-like patterns displayed different lengths and patterns because of the wide varieties of pulled proteins.

First the authors should describe better than “Processed traces were filtered and clustered according to ref.56, and their identification validated through the N2b tag as described (54)” in order for the reader to be able to repeat the experiment herein presented. Explain the steps with the specificities of the algorithm for TMEM16F.

Also, give the output: how many (absolute number, %, etc...) of curves were considered for each experiment.

Reply: We have now expanded the description of the filtering steps in the Methods. We used the code published with *Ilieva et al., 2020* (https://github.com/ninailieva/SMFS_clustering) and Fodis (<https://github.com/galvanetto/Fodis>). Specific parameters and detailed output numbers are now provided in the Methods section (SMFS unfolding experiments and analysis).

- One could also be concerned that the filtering might be affected by the variable topography and elasticity of the native membrane and as such by the uncertainty in the zero distance of the FD curve. Can you comment on this?

Reply: This is less of a concern: the unroofed cell membrane is certainly not atomically flat, but it has a roughness of less than 3 nm and a thickness of less than 10 nm. The cell membrane brakes when pushed at forces larger than 4 nN (see Galvanetto, 2018, <https://doi.org/10.1016/j.bbamem.2018.09.019>) so at much higher forces than the one used in our experiments. Therefore, we can expect an uncertainty in the position of the zero distance, but it is in the order of few nm — two orders of magnitude smaller than the size of the unfolding curves.

- Finally, and more importantly, there need to be a reasonable control experiment. This could for instance be a knock-out of the protein in native cell membranes. This could also be AFM-FS performed on a reconstituted system (supported lipid bilayer + protein). Or any other meaningful control experiment.

Reply: We thank the reviewer for this comment. To address this specific concern, which was also shared by the other referees (reviewer #1 - second comment, and reviewer #2 – second comment) we have performed additional unfolding experiments from purified TMEM16F reconstituted in artificial bilayer (soybean lipids). Although gleaning an adequate number of force-distance curve from this preparation turned out to be very challenging due to the low reconstitution yields, the data obtained recapitulated many of the features observed in TMEM16F unfolded from cell membrane fragments. The analysis of these new data indicates that the three studied constructs (endogenous, N-N2B- and reconstituted mTMEM16F (Fig. R1a)) have comparable unfolding patterns (Fig. R1b), variability of unfolding work (as evidenced by similar standard deviations values of 7.5, 8.4, and 8.7 aJ for N-

N2B-, native-, and reconstituted TMEM16F, respectively), and similar distributions with two dominant populations (Fig. R1c). We report these new data in Fig. 2 and Supplementary Figure S5 and detail the above thoughts in the Result section (pages 9,10)

Fig R1. (a) Sketches of the three constructs used in the SMFS experiments. (b) Density plots of $n=78$, $n=101$ and $n=68$ FD curves (left to right) of the constructs shown in a. (c) Violin plots of the calculated work to unfold the constructs in a. Average work is 16.4, 18.4, 20.4 aJ (left to right) with standard deviation of 7.5, 8.4 and 8.7 aJ, respectively. (The arrows highlight the presence of two dominant populations in all constructs, according to the unfolding work).

There are also few other minor points to address:

- Please explain in the text the added value of pulling from the inner membrane of an unroofed cell vs. pulling from the outer membrane of a living cell.

Reply: We thank the reviewer for allowing us to clarify this aspect. Pulling proteins requires a firm substrate for the protein that does not move when perturbed. Performing an AFM tip approach on a living cell at nN force range pushes the upper

cell membrane down and the vice versa happens during retraction. Therefore, the unfolding spectra of a membrane protein will be superimposed to an unpredictable change in offset due to the movements of the upper cell membrane, making the data interpretation not possible. In the revised version of the manuscript, we comment on this in the methods section.

- From ref. 56: “We should also underline that the method is not designed to distinguish different unfolding pathways of the same protein. The filtering and the clustering procedure are by far too coarse for this scope”. What about the fact that TMEM16F can be pulled by different subunits first, did you discard these curves?

Reply: This is an important aspect that was discussed in more general terms in *Galvanetto et al. (2022) Unfolding and identification of membrane proteins in situ. eLife 11:e77427* Fig. 1— supplement 7 and 8. It is possible that some curves in our dataset correspond to events when TMEM16F has been pulled by different subunits first, therefore generating shorter contour lengths. However, due to our filtering procedure, and in particular to the filtering based on total contour length (which is a very reliable observable in SMFS), shorter curves with a shorter contour length are not clustered together with the fully stretched TMEM16F.

- Fig 2c: did you test normality of the data to show a gaussian fit? The blue histogram in particular does not look normal. Maybe a box/violin plot is more adapted.

Reply: This is indeed a good suggestion. In the revised version we have replaced the histograms of work distribution with violin plots and removed the gaussian fits in panel 2c.

- Fig. 3e & video 1: there is no reason for this conformational change unless some energy is added to the system. Does this energy come from the AFM (tip scanning, laser temperature)?

Reply: Intramolecular vibrational dynamics under thermal equilibrium can be sufficient to overcome the energy barriers separating different conformers within the conformational space of proteins. Thermal fluctuations, for instance, drive the stochastic interconversion of ion channels between open and closed states, as evidenced by many decades of single-channel electrical measurements. The presence of multiple conformations under equilibrium conditions is particularly conspicuous in proteins exhibiting a shallow energy landscape, a scenario proposed for TMEM16F.

AFM imaging in amplitude-modulation mode is generally believed to be relatively gentle. A cantilever with a low spring constant was used in this study (0.1 Nm^{-1}), and extra care was taken to keep the free oscillation amplitude well controlled (approximately 2 nm) at set-points >90% of the free amplitude. Under these conditions, an energy of $1-3k_B T$ is estimated to be injected by the tip into the system. Such energy is usually not thought to perturb the operation of proteins, as it is

quickly dissipated and distributed among the many degrees of freedom of the molecule being investigated (<https://doi.org/10.1007%2Fs12551-017-0356-5>). However, we do not exclude that this small energy transfer might occasionally facilitate the interconversion between protein conformers.

- Please change "dozens" for the actual number of spectra.

Reply: We fixed this reporting the exact number.

- SMFS method: please mention how the InvOLS is calibrated, not just how the spring constant is.

Reply: We explained how we calibrated the InvOLS in the Methods section (according to the JPK Nanowizard 3 user manual).

- Fig. 4e & video 2: could this behaviour be a partial unfolding?

Reply: We do not believe that the observed change in lipid morphology around the TMEM16F is due to a partial unfolding or extraction of the protein from the membrane. The change in the bilayer structure extends over an area of up to a few hundred nm² whereas the free oscillation amplitude of the cantilever is only about ~2nm. Even if the molecule were accidentally picked-up by the AFM stylus only a few aa would be unfolded from the large NCD, given the modest vertical displacement of the probe. The tension transmitted to the transmembrane helices would therefore be negligible and a perturbation of the surrounding bilayer over areas of hundreds of nm² highly implausible.

- Fig. S2c: the glass coverslip is depicted like a tipless AFM cantilever, that is very misleading as a first read. Can you change it?

Reply: We have improved the figure to enhance its clarity.

Reviewers' Comments:

Reviewer #1:

Remarks to the Author:

I'm satisfied with the revision. I appreciate the authors' effort. I only have one minor comment. In fig 1 F & G, it would be helpful to add current scale bars in addition to their time scales.

Reviewer #2:

Remarks to the Author:

In my opinion, the authors appropriately addressed the referees' comments. The study, in its current form, represents a convincing and important contribution to the field.

Paolo Tammaro

Reviewer #3:

Remarks to the Author:

The authors have positively answered my concerns and as such I find the article to be suitable for publication.

To do so they have added significant control experiments and several methodological details that were previously lacking in the manuscript.

I am still skeptical about the answer about the membrane height variation and its effect on the determination of the zero distance: "The unroofed cell membrane is certainly not atomically flat, but it has a roughness of less than 3 nm," as one can see that some regions in the bilayer patch have clearly some higher topography than at the height profile location...

I don't see this comment in the methods section of the revised manuscript : "We thank the reviewer for allowing us to clarify this aspect. Pulling proteins requires a firm substrate for the protein that does not move when perturbed. Performing an AFM tip approach on a living cell at nN force range pushes the upper cell membrane down and the vice versa happens during retraction. Therefore, the unfolding spectra of a membrane protein will be superimposed to an unpredictable change in offset due to the movements of the upper cell membrane, making the data interpretation not possible. In the revised version of the manuscript, we comment on this in the methods section."

REVIEWERS' COMMENTS

Reviewer #1 (Remarks to the Author):

I'm satisfied with the revision. I appreciate the authors' effort. I only have one minor comment. In fig 1 F & G, it would be helpful to add current scale bars in addition to their time scales.

We thank the Referee for the careful feedback and overall positive reception of our work and revision. As suggested, we have added the scale bar (currents are normalized to the maximum current in panels 1f,g).

Reviewer #2 (Remarks to the Author):

In my opinion, the authors appropriately addressed the referees' comments. The study, in its current form, represents a convincing and important contribution to the field.

Paolo Tammaro

We thank Prof. Paolo Tammaro for the reception of our revision and the very constructive comments provided throughout the revision process.

Reviewer #3 (Remarks to the Author):

The authors have positively answered my concerns and as such I find the article to be suitable for publication.

To do so they have added significant control experiments and several methodological details that were previously lacking in the manuscript.

We thank the Reviewer for the positive reception of our revision and helpful suggestions.

I am still skeptical about the answer about the membrane height variation and its effect on the determination of the zero distance: "The unroofed cell membrane is certainly not atomically flat, but it has a roughness of less than 3 nm," as one can see that some regions in the bilayer patch have clearly some higher topography than at the height profile location...

The reviewer is correct; protrusions on the lipid bilayer larger than 3 nm indeed do occur, despite the average roughness being below ~3 nm. However, we would like to stress that these corrugations are certainly more than an order of magnitude smaller than the fully stretched channel (>300 nm). In addition, we only performed SMFS on the flattest areas of the membrane patches to avoid larger debris. Consequently, we expect that their impact on our contour length determination is still relatively minor. In the revised method section "SMFS unfolding experiments and analysis", we acknowledge such corrugations and briefly comment on the potential effect on the filtering procedure.

"We only performed SMFS on the flattest areas of the membrane patches to limit uncertainties arising from membrane corrugations and height variation (Fig. 1c) on this filtering step."

I don't see this comment in the methods section of the revised manuscript : "We thank the reviewer for allowing us to clarify this aspect. Pulling proteins requires a firm substrate for the protein that does not move when perturbed. Performing an AFM tip approach on a living cell at nN force range pushes the upper cell membrane down and the vice versa happens during retraction. Therefore, the unfolding spectra of a membrane protein will be superimposed to an unpredictable change in offset due to the movements of the upper cell membrane, making the data interpretation not possible. In the revised version of the manuscript, we comment on this in the methods section."

We apologise, we have now added a comment at the end of the section: SMFS unfolding experiments and analysis

"This method is inherently incompatible with unfolding proteins with both C- and N-termini on the extracellular side of the membrane. One solution could be to perform an AFM tip approach on a living cell at nN force range. However, during retraction, the upper cell membrane is pushed down a few hundred nm and vice versa. The unfolding spectra of a membrane protein will therefore be superimposed by an unpredictable change in offset due to the movements of the upper cell membrane, making accurate data interpretation difficult."